

# Factors influencing lake surface water temperature variability in West Greenland and the role of the ice sheet

Laura Carrea[1,*], Christopher J. Merchant[1,2,*], Richard I. Woolway[3], and Niall McCarroll[1,2]

[1]Meteorology Department, University of Reading, Reading, United Kingdom
[2]National Centre for Earth Observations, University of Reading, Reading, United Kingdom
[3]School of Ocean Sciences, Bangor University, Bangor, United Kingdom
[*]These authors contributed equally to this work.

**Correspondence:** Laura Carrea (l.carrea@reading.ac.uk)

**Abstract.** Subarctic West Greenland is populated by thousands of seasonally ice-free lakes. Using remotely sensed observations, we analyse the surface water temperatures of six lakes during 1995-2022 to identify factors influencing their variability. The connectivity to the Greenland Ice Sheet (GrIS) has a clear influence on lake surface temperature, with ice-sheet marginal lakes experiencing smaller average summer maximum temperature ($< 6°C$) and minimal inter-annual variability. A lake fed by

a GrIS-originating river has the fastest seasonal response and largest seasonal amplitude with average maximum temperatures above $13°C$. The seasonal cycle of surface water temperature for all studied lakes is asymmetrical, with faster warming observed after ice off, and a slower cooling of water towards winter freezing. We find that during the study period, the onset of positive stratification has occurred earlier, at rates of up to 0.5 days year$^{-1}$, and that July-August temperatures have increased at rates up to 0.1°C year$^{-1}$, although the GrIS-connected lakes show smaller increases. Our analysis suggests that the main me-

teorological factor determining interannual variability of surface water temperature in the studied lakes is air temperature. This study highlights the important role of remote sensing for long-term monitoring of Greenlandic lakes under climate change.

## 1   Introduction

The subarctic expanse of West Greenland is home to a multitude of lakes (How et al., 2021), their seasonal ice-free periods providing a window into the region's dynamic environmental processes (Anderson et al., 2017; Fowler et al., 2020). Influenced

by climatic and glacial factors, these water bodies offer invaluable insights into the intricate interplay of climatic forces shaping high-latitude ecosystems (Saros et al., 2019; Prater et al., 2022; Saros et al., 2023). Among the myriad consequences of climate change on these lakes, one of the most pressing concerns is the direct and indirect impacts of rising lake surface water temperature (LSWT). This phenomenon not only affects various ecological states, structures, and processes (Williamson et al., 2009; Woolway et al., 2022) but also holds implications for ice sheet mass balance (Zhang and Bolch, 2023). Specifically, the

interaction between LSWT and ice margin dynamics can accelerate glacier mass loss, particularly in areas where lakes abut ice fronts. Studies suggest that such lakes contribute to enhanced thermo-mechanical erosion and reduce basal resistance, resulting in heightened ice flux towards the margins (King et al., 2019; Maurer et al., 2019; King et al., 2020; Benn et al., 2007; Tsutaki et al., 2011; Pronk et al., 2021). Given the ongoing trajectory of global climate change, understanding the factors driving



LSWT variability in Greenland is paramount for understanding broader environmental shifts. Traditionally, global studies investigating variations in LSWT have relied on in situ measurements, yet such data are notably sparse in West Greenland (Kettle et al., 2004). Indeed, the logistical costs and monitoring challenges in this remote region, especially over decadal time scales, have hindered comprehensive research efforts. Consequently, most lakes in West Greenland remain largely unexplored, notwithstanding an extensive investigation of more than 80 lakes mainly located along a transect between the GrIS and the coast south of Sisimiut, but with no hydrological connection to the GrIS (Anderson et al., 2001). The absence of widespread, long-term monitoring has left the thermal dynamics of many lakes poorly quantified, impeding accurate estimations of their role in regional energy and mass exchange with the atmosphere and interactions with the GrIS margins, particularly in the context of climate change impacts (Anderson et al., 2001). However, recent advancements in remote sensing have transformed our ability to observe environmental variables across vast spatial scales and extended time frames. Leveraging these capabilities, our study investigates the thermal behaviors of six representative lakes (Figure 1) across West Greenland using satellite-derived time series data of LSWT and Ice Cover (LIC). Spanning from 1995 to 2022, our study explores the intricate dynamics of these aquatic ecosystems in response to climatic and glacial influences. These six lakes were carefully chosen to encapsulate the diverse environmental conditions prevalent in West Greenland. Notably, two lakes are directly linked to the GrIS, serving as sentinel sites for evaluating the impacts of glacial processes on lake thermal regimes (see Methods). Additionally, one lake, indirectly connected via a river fed by the GrIS, provides valuable insights into the downstream ramifications of glacial meltwater discharge on aquatic systems. Our investigation extends beyond simple characterization of seasonal temperature fluctuations by exploring the complex interrelationships between climatic variables and lake dynamics. By analyzing LSWT and LIC data alongside concurrent meteorological parameters such as air temperature and shortwave radiation, we investigate the underlying mechanisms governing thermal variability in these lakes. We have included in our study lakes that connected to the GrIS, as well as those that are not, in order to gain insights into the influence of the ice sheet on lake thermal properties. Central to our investigation is the detection of climatic signals imprinted within the temporal evolution of LSWT, particularly against the backdrop of observed global warming trends. By examining the inter-annual variability and trends in surface temperature, we aim to investigate the extent to which climatic warming influences the thermal regimes of these high latitude lakes. Furthermore, by scrutinizing the phenology of warm-stratified conditions, we aim to discern alterations in the seasonal dynamics of these aquatic environments.

## 2 Data and Methods

### 2.1 Study region

The west of Greenland has its most substantial ice-sheet free margin between 63.5°N and 68.5°N. The seasonally ice-and-snow-free parts of this region generally have an elevation between 0 and 1100 m above sea level (masl). The edge of the GrIS is mostly found at elevations ranging from ~600 m to ~1200 m, and at distances up to 180 km from the west coast (excluding fjords). The intervening area is characterised by thousands of inland water bodies exhibiting a range of limnological conditions (Ryves et al., 2002). Specifically, lakes in this region vary in area, depth, elevation, components of water balance (groundwater,



precipitation, surface runoff and stream feeding), surface connectivity to other lakes via streams (which may be seasonal), chemistry, connectivity to the ice-sheet margin and water clarity. The latter two aspects are linked, as glacial meltwater contains fine particulates in suspension, particularly minerogenic glacial flour (Burpee et al., 2018). The particulates cause high turbidity,

with associated ecosystem impacts through reduction in photosynthetically active radiation (Sommaruga, 2015). Turbid lakes may respond not only directly to climatic changes in weather-driven forcing, but also indirectly via modification of albedo associated with changing glacial fluxes, since water clarity directly influences lake heat budgets and stratification (Saros et al., 2019). The climate of the study region also varies considerably, principally from the coast to the area next to the GrIS (Anderson et al., 2001), with consequent changes in vegetation and lake properties. Areas close to the GrIS have a Low Arctic continental

climate, with mean annual surface air temperature of -6°C, an annual temperature range of ∼30°C, continuous permafrost, and low precipitation ≪ 150 mm year$^{-1}$. The zone from the GrIS to mid-way towards the coast is characterized by very low (negative) effective precipitation. The coast is characterised by a Low Arctic maritime climate, with a lower annual temperature range (∼25°C) and higher precipitation (300 mm year$^{-1}$). The moderated summer maximum temperatures and the presence of coastal fog banks associated with the more maritime climate imply a longer presence of snow and snow packs in this region

(Anderson et al., 2001). In this study, the thermal behaviour of six relatively large lakes from 1995 to 2022 is investigated using satellite-derived time series of LSWT and LIC from the European Space Agency Climate Change Initiative (ESA CCI) (Hollman et al., 2013) project dedicated to lakes (Carrea et al., 2023) - see below. The six lakes have diverse properties that are representative of those distributed throughout West Greenland. Two lakes are directly connected to the GrIS (they are glacier terminal lakes), and one lake is indirectly connected by a river of length ∼92 km fed by the GrIS. These three lakes are at

elevations of 300 m or higher. The other three lakes are unconnected to the GrIS and at elevations less than 50 m, with one very close to the coast (Figure 1).

## 2.2 Digital elevation model

To explore the topography of the study region, and that near the vicinity of the studied lakes, we used the Terra Advanced Space-borne Thermal Emission and Reflection Radiometer (ASTER) Global Digital Elevation Model (GDEM) Version 3 (ASTGTM)

(NASA/METI/AIST/Japan Spacesystems and U.S./Japan ASTER Science Team, 2019; Abrams et al., 2020), jointly created and developed by National Aeronautics and Space Administration (NASA) and Japan's Ministry of Economy, Trade, and Industry (METI), provides a global digital elevation model (DEM) of land surface on Earth that extends from 83°N to 83°S, has at a horizontal spatial resolution of 1 arc second (about 30 meter at the equator). The data are projected onto the 1984 World Geodetic System (WGS84)/1996 Earth Gravitational Model (EGM96) geoid. This dataset can be downloaded at the

Land Processes Distributed Active Archive Center (LP DAAC) https://doi.org/10.5067/ASTER/ASTGTM.003.

## 2.3 Lake bathymetry

Bathymetry data (maximum depth, mean depth, area and volume) for each of the studied lakes were extracted from the GLO-Bathy dataset (Khazaei et al., 2022) to characterise the lakes in the region. GLoBathy provides estimates of bathymetry for



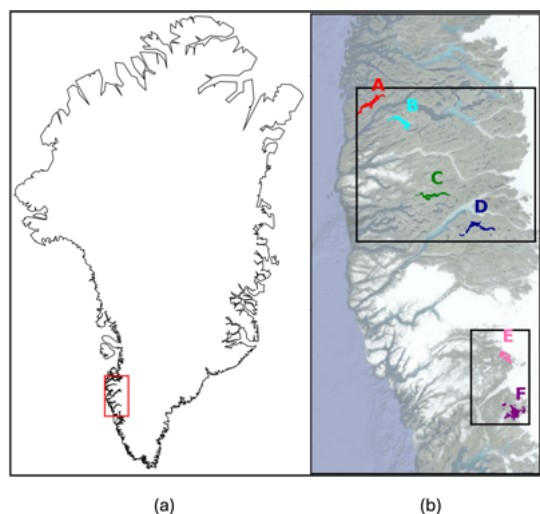

**Figure 1.** (a) Map of Greenland showing the general location of the main western ice-free margin. (b) Map of the western ice-free margin, showing the north and south study domains (boxes) and (colour-coded) locations of six major lakes for which temperature time-series are available.

lakes worldwide with reasonable accuracy, given the complexity of estimating underwater topography. The dataset is available

at *figshare* https://doi.org/10.6084/m9.figshare.c.5243309.v1.

## 2.4 Landsat 8

To characterise the studied region and the spatial variability of the lakes in the region, we analysed data from the Thermal Infrared Sensor (TIRS) and the Operational Land Imager (OLI) instruments on Landsat 8 (www.usgs.gov). The two sensors provide a spatial resolution of 30 meters for the visible, near-infrared (NIR), short-wavelength infrared (SWIR) and of 100

meters for the thermal channels. The repeat coverage is about 16 days and the scenes are provided by the Worldwide Reference System-2 (WRS-2) path/row system, the size of which is 185 km x 180 km, and with a swath overlap percentage increasing from the equator to extreme latitudes (Vanhellemont and Ruddick, 2015). The Landsat 8 data are available at the Landsat Data Access web page (https://www.usgs.gov/landsat-missions/landsat-data-access) where to discover how to search and download all Landsat products from United States Geological Survey (USGS) data portals. The USGS Landsat no-cost open access data

policy remains intact since its inception in 2008.

## 2.5 ESA CCI Lakes

For most of this work, we utilized LSWT and LIC from the European Space Agency (ESA) Climate Change Initiative (CCI) Lakes dataset v2.1 (Carrea et al., 2024) and V2.0.2 (Carrea et al., 2022a) respectevely (Carrea et al., 2023). LSWT observations are derived from the following visible and infrared radiometers on polar orbiting satellites: ATSR-2 (1995–2002), AATSR



(2002–2012), MODIS on Terra (2000-2022), AVHRR on MetOpA (2007-2019), AVHRR on MetOpB (2012-2019), SLSTR
        on Sentinel3A (2016-2022), and SLSTR on Sentinel3B (2018-2022). The LIC observations are derived from MODIS on Terra
        and Aqua (2000-2020). The LSWT and LIC have been retrieved at a spatial resolution of ∼1 km at nadir and then gridded
        to a common grid 1/120° cells on a global lake mask at 1/120° resolution (Carrea et al., 2023). LSWT are accompanied by
        per-pixel quality levels which summarise the confidence in the retrieval. Only LSWT of quality 4 and 5 (the highest) have been
used for this study. Given that the best resolution of the instruments used for the retrieval of the LSWT is 1 km, obtaining
        reliable information for each of the six lakes is challenging as their maximum distance to land is between 1.2 km and 1.8 km.
        In these cases, the LSWT retrieval is very likely to be available only for that part of the lake and only for a limited proportion of
        almost daily overpasses (clear sky and observations relatively central within the swath). The LIC is retrieved using reflectance
        channels, therefore the illumination at a local time of ∼11 am becomes marginal or absent from November until the middle
of January. The LSWT and LIC of the ESA CCI Lakes dataset have been validated through comparison with in situ data and
        through manual inspection respectively (Carrea et al., 2023). Because the LSWT retrieval algorithm is based on physics, a
        stable performance is expected across domains in times and spaces where LSWT cannot be directly easily validated. This is
        the first consistent long term LSWT and LIC time-series from satellite for lakes in Greenland. The ESA CCI Lakes v2.0.2 and
        v2.1 dataset are available for download at the Centre for Environmental Data Analysis (CEDA - United Kingdom) archive.

**2.6 ESA CCI Clouds**

        The bottom of atmosphere downwelling shortwave radiation from the ESA CCI Clouds dataset (Stengel et al., 2020) has been
        used in this study for establishing if any correlation between LSWT and shortwave solar radiation is present. The dataset is
        available at https://dx.doi.org/10.5676/DWD/ESA_Cloud_cci/AVHRR-AM/V003.

        **2.7 ESA CCI Sea Surface Temperature**

The sea surface temperature dataset used in this paper to detemine the marine influence on the lake close to the coast. We
        use the ESA CCI SST dataset (Embury et al., 2024) and it can be downloaded at the Centre for Environmental Data Analysis
        (CEDA - United Kingdom) archive http://dx.doi.org/10.5285/62c0f97b1eac4e0197a674870afe1ee6. The dataset utilised for
        this work is monthly L4 SST (analysis data) from 1995 to 2021. The monthly aggregated datatset can be downloaded at the
        SST regridding service of the University of Reading (https://surftemp.net/regridding/index.html).

**2.8 ECWMF ERA5-Land data**

        Data for two atmospheric variables, 2m air temperature and solar radiation (surface solar radiation downwards), used in this
        study are from the ERA5-Land reanalysis dataset of the European Centre for Medium-Range Weather Forecasts (ECMWF)
        (Hersbach et al., 2020; Muñoz-Sabater, 2019). Data at three hourly intervals have been used to create a climatology and
        monthly values. The values of 2m air temperature have been corrected because of the difference in the nominal altitude of
the ERA5-Land dataset and the high-resolution DEM from ASTER. ERA5-Land is a high-resolution version of the land com-





ponent of the ERA5 reanalysis (Hersbach et al., 2020). It is run at 9 km resolution, compared to ERA5 at 31 km, before being interpolated onto a regular 0.1° lat/lon grid. This allows for representation of smaller-scale variability than in ERA5 and other reanalysis products. However, there are important limitations to ERA5-Land. Only the land component of the ECMWF Integrated Forecasting System (IFS) is run, with interpolated ERA5 fields providing forcing from other components (i.e. atmo-

sphere, ocean) without two-way coupling. Variables such as surface pressure, wind, precipitation and radiative fluxes therefore do not capture variability at sub-31 km scales, despite being available on the ERA5-Land grid. There is no data assimilation, although observations do indirectly affect the simulation via the forcing from ERA5. The dataset can be downloaded at the Copernicus Climate Data Store https://doi.org/10.24381/cds.e2161bac.

## 2.9    Analysis

### 2.9.1    Characterisation of lakes in the study region

To verify that the six studied lakes are representative of most lakes in the study region, we quantified the physical character-istics of observable lakes in the region using satellite instruments on Landsat 8. We found that most lakes are smaller than those detectable with meteorological satellite sensors (used for the ESA CCI Lakes dataset), which provide quality data and offer frequent revisit time (~2 days) but have a spatial resolution of (nominally) 1000 m. Consequently, to explore the spatial

variability of the lakes in the region, we utilised higher-resolution remote sensing observations from Landsat 8. Even with Landsat 8's finer thermal resolution (nominally ~100 m), careful measures were taken to minimize the influence of mixed land-lake pixels around lake peripheries on the derived lake-averaged properties. For this analysis, only lakes with an area exceeding 2.5 km$^2$ were characterised using Landsat 8 thermal observations. The characterisation involved maximum bright-ness temperature, elevation, water albedo and lake area. The elevation and a limited (in time) ultra-clear-sky sample of visible

reflectance and infrared brightness temperature data from 2015 and 2021 (see Data) were resampled onto a common 50 m-resolution latitude-longitude rectilinear grid to resolve smaller lakes. A composite image was created from several clear-sky Landsat 8 images taken between 2015 and 2021. This composite image consisted of the time-minimum reflectance in the red, blue and green channels in these images. To identify pixels with liquid water, the modified normalised difference water index (NDWI) (Yang and Du, 2017; McFeeters, 2021; Xu, 2006) was applied to the minimum reflectance composite. Shadows north

of cliffs were additionally filtered out from the resulting inland water mask using the slope of the ASTER DEM. The DEM is also used to compare the thermal/ice characteristics of the study region to elevation. The area of the lakes was estimated using the water-filled pixels derived from the modified NDWI. Water albedo was estimated by adapting method 1 of Liang (Liang, 2001) to the corresponding bands of Landsat 8. Finally, for each selected lake, we calculated the following properties: lake-median elevation, lake-mean time-maximum temperature, lake-mean time-minimum albedo and lake area. The statistical

estimator used for each property was chosen based on its appropriateness to the property and the nature of possible outliers. For example, taking the minimum across the summer Landsat samples ("time-minimum") of albedo ensures it represents the albedo of the ice-free water body, avoiding skewing of the results by any samples where the lake is only partially ice free.





### 2.9.2 Determination of the climatological curves

The climatology of all variables in this study, except for LIC, has been calculated for each day of the year using a 15-day
smoothing window (7 days on either side of the central day). This analysis is based on data collected from 1995 to 2020.
Additionally, the standard deviation and the number of observations have been evaluated to assess the variability of temperature
on specific days across the years, with particular focus on the number of observations for LSWT and LIC. The LIC climatology
has been computed as the percentage of ice relative to the total number of ice or water observations, based on data from 2000 to
2020. This climatology is accompanied by the number of days when ice was observed. To reduce false positives, only days with
at least 50 observations and confirmed ice presence were included in the LIC climatology. The climatological curves for LSWT
and LIC have been derived from the ESA CCI Lakes dataset and the curves for air temperature and the solar downwelling flux
have been derived from the ERA5 Land dataset.

### 2.9.3 A proxy for stratification phenology

Using observations for LSWT and LIC (ESA CCI Lakes dataset), inferences about the mixing state of lakes can be made
(Woolway et al., 2021). In this study, we use LSWT for estimating lake stratification phenology, assuming that the six lakes
are dimictic (or monomictic) such as most of the lakes in southwest Greenland (Saros et al., 2016). The day of year in which
LSWT crosses the threshold of 3.98°C, the temperature of maximum density of freshwater, is taken as a phenological indicator
of stratification (Woolway et al., 2021; Fichot et al., 2019). Where consecutive days with observations are not available, the
available days are linearly interpolated in time before the threshold-crossing day is calculated.

### 185 2.9.4 Trend and correlation anaylsis

Trends have been calculated using a linear regression model and the standard uncertainty are reported. The correlations are
evaluated as Spearman correlations, assuming a monotonic relationship and statistically significant correlations (with a p-
value<0.05) are highlighted. The relation between LSWT and air temperature is evaluated as a function of time to investigate a
potential lag between air temperature and LSWT. The LSWT monthly mean for each of the ice-free months is correlated with
the monthly air temperature values for all the months of the year.

## 3 Results

### 3.1 Physical characterisation of the study region

In this study, we first characterise the physical environment of the studied lakes in terms of the landscape in which they are
located, as well as their elevation, surface reflectance and surface temperature variability. Two domains are addressed: (i)
northern domain, which includes four of the six target lakes, and (ii) the southern domain, which includes two lakes (Figure
1). The colour coding of these water bodies in Figure1(b) are used throughout the results section.





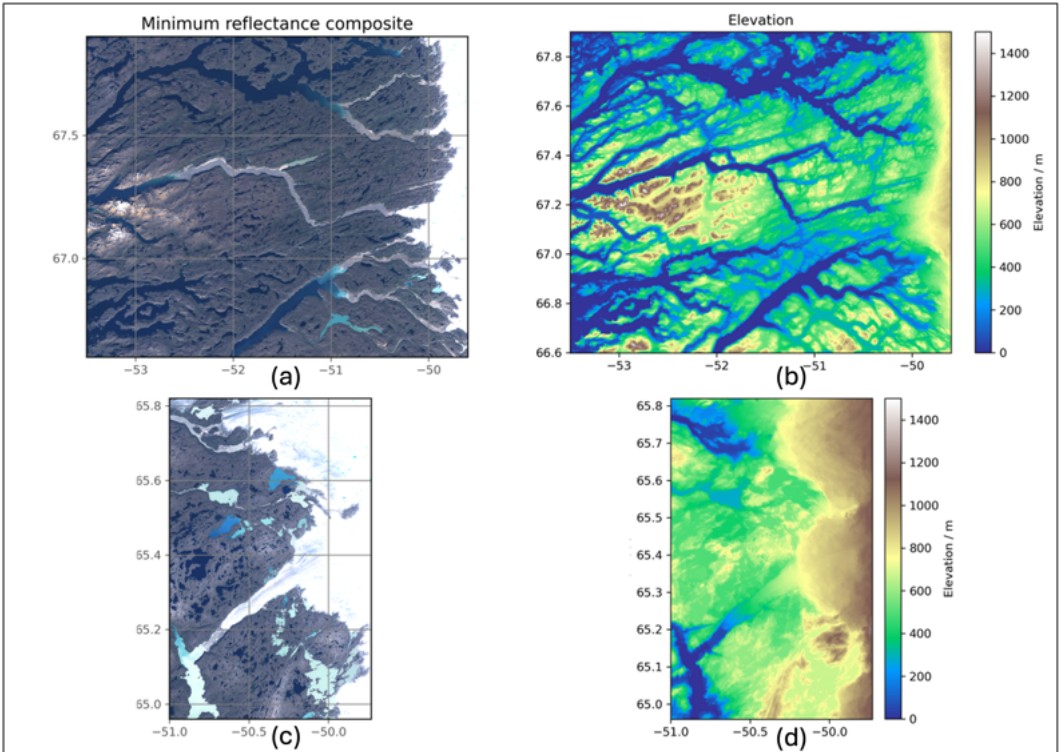

**Figure 2.** (a) Natural-colour composite of the minimum reflectance in red, green and blue bands of Landsat 8, northern domain. (b) Elevation from ASTER, northern domain. (c) and (d), as (a) and (b) for southern domain.

Figures 2(a) and 2(c) show the clear-sky appearance of the northern and southern study areas respectively in summer at a time of minimum snow and ice cover. The image is composited from several clear-sky Landsat 8 images occurring between 2015 and 2021 inclusively (see Data and Methods). Comparing thermal/ice characteristics (Figure 2(a) and 2(c)) to elevation (Figure 2(b) and 2(d)), we find that ice and snow persist throughout summer at elevations exceeding ∼1200 m (mountains to west in the northern domain) while the ice-sheet margin is seen at elevations from ∼600 m. The modal elevation of the domain is ∼400 m. Many dark lakes are visible across the landscape of seasonally snow-free permafrost, as well as several prominent bright-blue ice-free water bodies (most obviously in the southern domain) (Figure 2). The latter are directly adjacent to glacier termini or are connected to glacial water inputs by streams or river. Lakes situated in the northern domain are generally more elongated, as is typical of lakes formed in glacially eroded valleys. The lakes of the southern domain more often represent flooded basins in the landscape, sometimes joined by rivers. Using Landsat 8 thermal observations, we identify 90 lakes in the northern domain and 48 in the southern. These include the six lakes for which timeseries data (from the ESA CCI Lakes) are available.

Figure 3 shows the minimum albedo and, in both domains, the higher albedo lakes are either directly connected to the ice sheet or connected to other bright lakes by rivers (compare Figures 2 and 3). The physical properties of these 138 water bodies of





both domains are shown in Figure 4. Most (79%) lakes are dark with albedo of <5%. Of these, 86% were observed with a lake-mean time-maximum temperature exceeding 287.15 K (14°C); however, these warmer lakes tend to be smaller, representing only 66% of the low-albedo water surface area. There is no apparent role for elevation in determining the maximum observed LSWT. This is because higher air temperatures can occur across the central band of the margin at middle elevations, which

are maximally distant from the temperature-moderating effects of either the sea or the ice-sheet. High albedo lakes are not observed at the lowest elevations (<100 m) but occur across the full range of lake surface areas. High-albedo lakes tend to attain less extreme maximum temperatures. This can be partly explained by these lakes reflecting more incoming solar irradiance. Additionally, the inflow of meltwater from the ice-sheet (including indirectly via rivers), which correlates with higher albedo, is likely a contributing factor. Meltwaters initially have a temperature just above freezing and likely remain cooler than the

lake temperature when they reach the lakes. This influx of colder water corresponds to a negative heat flux to the lake energy balance. The six target lakes with time-series data mostly lie within the distribution of other lakes in the landscape, except for lake A. Lake A has low albedo (indicating good solar absorption), yet its maximum temperature in the Landsat 8 observations is only 281 K ($\sim$ 8°C). This particularly low temperature is partly an artefact of temporal sampling, as the corner of the domain where Lake A is located was not observed on the same day that the lakes in the rest of the domain were observed at their

maximum temperature. Nonetheless, the general maritime climate of Lake A's location may also play a role.

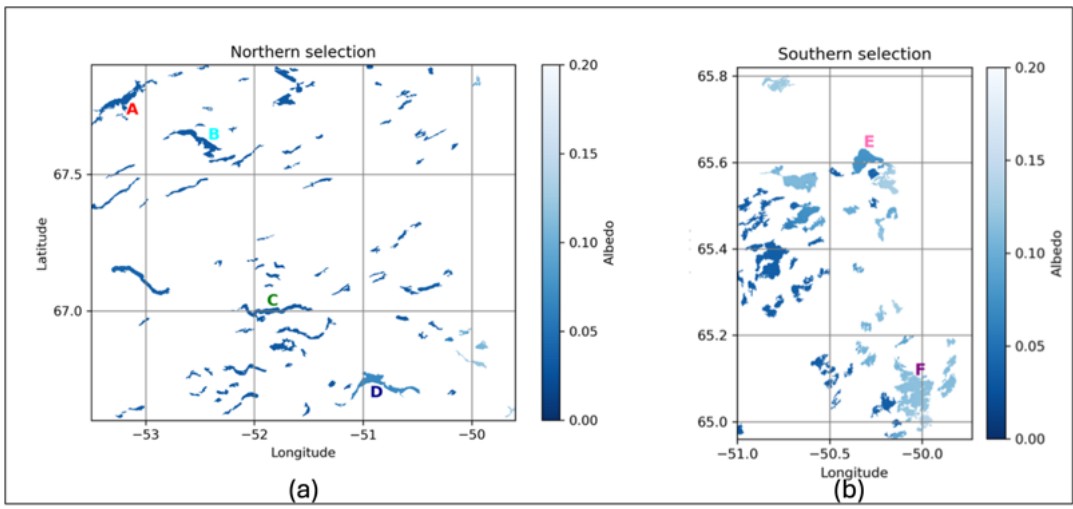

**Figure 3.** Lake pixels selected for remote sensing of albedo and temperature characteristics in the (a) northern and (b) southern domains. The lake pixels are coloured according to their lake-mean minimum albedo in the Landsat 8 sample (seeData and Methods).

## 3.2 Characterisation of the six studied lakes

The six studied lakes (Table 1) in west Greenland represent the variety of larger lakes in the region. Each of the six studied lakes have outflows that at least seasonally connect to fjords or the coastal sea. All the six lakes have maximum distance from



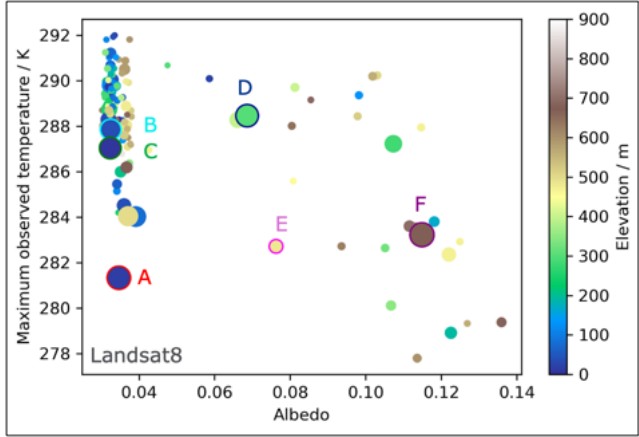

**Figure 4.** Distribution of physical characteristics of remotely sensible lakes across both study domains. The areas of the scatter points are in proportion to the lake areas. The points corresponding to the six lakes with time-series data are distinguished by borders in accordance with the colour scheme of Figure 1(b).

the closest land (Carrea et al., 2015) less than 2 km which is a useful measure for the size of the lakes in the context of LSWT

remote sensing. Consequently, the six lakes can be resolved in the satellite imagery, given that the best nadir resolution of the meteorological instruments used for LSWT retrieval is 1 km. However, observing these six lakes remain challenging.

Lake E and F are situated very close to the GrIS, characterised by a Low Artic continental climate and directly connected to the ice sheet (Figure 2(c)). They have high albedo due to the presence of fine particulates in the meltwater deriving from the GrIS. While lake F shows a relatively uniform spatial distribution of the albedo, lake E is composed of three sub-basins with

very different albedos (0.12, 0.08 and 0.04), suggesting the inter-basin mixing and flows are weak and/or episodic (depending on lake water levels). Nonetheless, for the purpose of this study, we consider it as a single lake. Most of the LSWT observations for Lake E are from the northern sub-basin where the albedo is 0.08, similar to that of lake D. Figure 5 shows the number of observations across the time-series for each of the cells of lake E present in a raster of inland water locations used to assist the identification of water-only pixels associated with the lake in meteorological satellite processing (Carrea et al., 2022b).

It illustrates that the lake mean temperature obtained is dominated by the centre of the largest sub-basin away from the lake margins where pixels are often detected as partially overlapping land, and therefore not used for LSWT. Some narrow portions of lake E are never observed with the meteorological sensors for LSWT, which is also true of the other lakes. Lakes E and F are, among the six studied lakes, situated at the highest elevation, and at the lowest latitudes. Lake F has the largest area and is most likely the deepest lake according to GLOBathy. Lake D is located about 50 km (directly) from the GrIS with an albedo

close to the one of lake E. Lake D is connected to the ice sheet by a ~92 km long stream. This stream transports particulates from small glacier-terminal lakes at an elevation of about 900 m to lake D, at an elevation of about 300 m. We expect that this distance is sufficient for the water from the melting ice sheet to be modified, in terms of its temperature, before reaching lake D, despite the albedo of lake D demonstrating that significant glacial flour remains suspended and is transported to the lake.





**Table 1.** List of the six lakes present in the ESA CCI Lake dataset and their characteristics.

| CCI Lake identifier | 300002493 | 300002545 | 300002685 | 2199 | 3219 | 1099 |
|---|---|---|---|---|---|---|
| **Identifier in this paper** | A (red) | B (cyan) | C (green) | D (blue) | E (pink) | F (purple) |
| **Latitude lake centre** | 67.779 | 67.607 | 66.999 | 66.754 | 65.615 | 65.087 |
| **Longitude lake centre** | -53.179 | -52.429 | -51.887 | -50.937 | -50.318 | -50.040 |
| **Maximum distance to land (km)** | 1.6 | 1.4 | 1.2 | 1.8 | 1.8 | 1.8 |
| **Max depth (m)** | 95 | 38 | 44 | 28 | 23 | 302 |
| **Mean depth (m)** | 27 | 11 | 13 | 8 | 6 | 59 |
| **Volume (km$^3$)** | 2.0 | 0.7 | 0.9 | 0.6 | 0.3 | 10.0 |
| **Area$^a$ (km$^2$)** | 76 | 64 | 65 | 82 | 55 | 170 |
| **Area L8$^b$ (km$^2$)** | 68 | 54 | 58 | 63 | 48 | 72 |
| **Albedo L8$^c$** | 0.03 | 0.03 | 0.03 | 0.07 | 0.08 | 0.11 |
| **Elevation DEM$^d$ (m)** | 15 | 36 | 5 | 299 | 479 | 671 |
| **NWP nominal elevation$^e$ (m)** | 150 | 335 | 318 | 534 | 590 | 904 |

[a] From GLOBathy Khazaei et al. (2022)

[b] Sensible area, Landsat8

[c] Derived from Landsat8

[d] From ASTERAbrams et al. (2020); NASA/METI/AIST/Japan Spacesystems and U.S./Japan ASTER Science Team (2019)

[e] From ERA5-LandHersbach et al. (2020); Muñoz-Sabater (2019)

Lakes A, B and C absorb a greater proportion of insolation than D, E or F, having low albedo. They are also at lower elevation.
Lake A is the lake located at the highest latitude, the lowest elevation, and nearest to open sea. The maximum depth for lake A, as reported by the GLOBathy database, is 95 m, but this may be an over-estimate for a lake so close to the sea and at an elevation of 15 m. Lake B is further inland, separated from Lake A by a fjord and mountains reaching a height of about 600 m. Lake C is roughly midway between open sea and the GrIS.

## 3.3 Seasonality of LSWT and LIC

The mean seasonal cycles of the LSWT and LIC are driven by the prevailing climatology of relevant meteorological factors, such as air temperature and insolation. Changes in the seasonal cycle may occur in interaction with climate change.

### 3.3.1 Relation between LIC and LSWT seasonality

All the six studied lakes have at least partial ice-free water surfaces in May (the timing of which differs between lakes) to at least mid-November (Figure 6). Figure 6(a) shows the climatology for the centre location of the lakes reported in Table 1. LSWTs are retrieved through to the middle of November, by which time all lakes have cooled below 4°C, such that water column mixing has occurred. All six lakes have complete ice cover at the lake centre for a period starting after mid-November




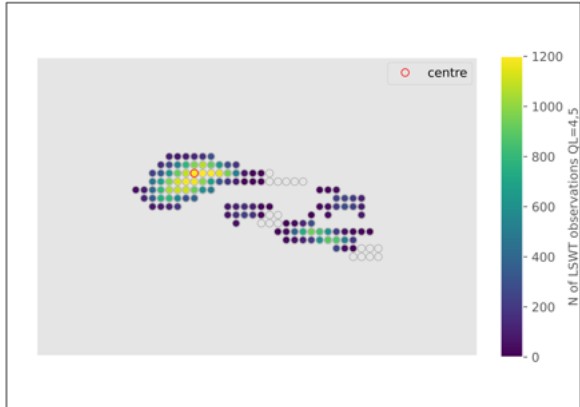

**Figure 5.** Number of observations for each $1/120°$ pixel from 1995 to 2022 for lake E.

(timing not observed due to no daylight) and ending in May, Figure 6(b). However, the timing of full freezing (on average) of the water surface is uncertain in this dataset because of the observational limits LIC from November to mid-January due to marginal or absent during this period (see Data section). An increase of ice presence can be observed already from mid-October

for the lakes directly or indirectly connected to the GrIS (D, E and F), and it can be assumed that these lakes fully freeze shortly after the last LSWT measurements are available. During summer, the reported LIC is less than 0.2% but greater than zero for all the lakes, even at time of maximum LSWT. This is thought to reflect a base level of misclassification (false ice detection) in these observations rather than ice being present, although, in the case of lakes E and F, there might be some detection of ice detached from the GrIS (Mallalieu et al., 2020). An unambiguous feature is that lakes E and F have later and slower loss of

ice cover during May and June. The earliest ice breakup (on average) happens for lake C, which is the lake with the smallest maximum distance to land (Table 1). For lakes A, B and C, the seasonal cycle of the ice cover (Figure 6(b) shows that the percentage of ice pixels is still low by mid-November. LSWTs may therefore also still be obtained in November for ice-free parts of the lakes.

### 3.3.2   Maximum average temperatures

The climatological peak temperatures are in broad correlation with the maximum temperature from Landsat 8 reported in Figure 4. Each of the six lakes warm, on average, to a temperature greater than $4°C$. The maximum depths estimated for the studied lakes make it plausible they are stratified during periods when LSWT exceeds $4°C$ although potential stratification is only brief in the case of lake E and F. As the studied lakes are covered with ice for approximately half of the year, they can most likely be characterised as dimictic. The coolest studied lakes are E and F, which are not only situated at the highest elevation

but are also directly influenced by the GrIS meltwater. Lakes E and F reach their climatological maximum value of LSWT of $6.5\pm1.6°C$ and $7.5\pm1.7°C$ on the 14th and 15th of August respectively. Between May and the end of August, LSWT in lake F is between $1°C$ and $2°C$ warmer on average than lake E. Lake E is likely colder than F due to its smaller size and shallower depth, and perhaps due to it being more easily cooled by the melt water from the GrIS. However, the two lakes have very



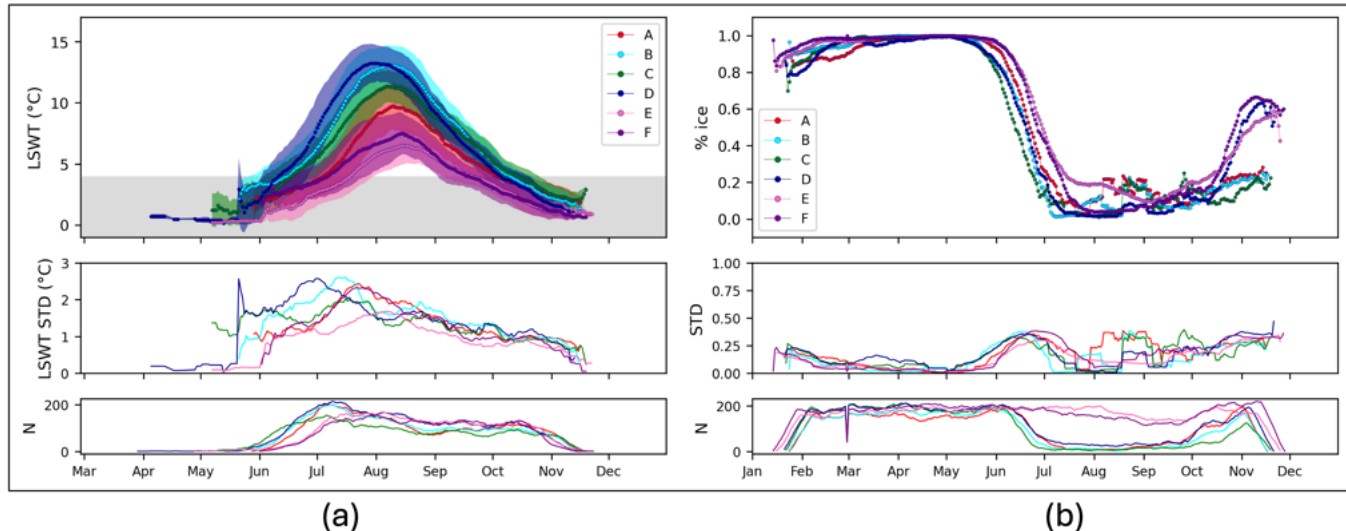

**Figure 6.** (a) Seasonal cycle of LSWT, calculated as day-of-year mean over 1995 to 2020 of a centred 7-day aggregation window temperatures observations at the lake centre location in kelvin; the shaded area represents one standard deviation away from the mean (upper panel) and the grey shaded area features temperatures below 4°C. (b) Seasonal cycle of LIC as the percent of cells covered by ice with respect to the total number of valid (ice and water) observations on the lake. Only cases where the number of observations were greater than 50 (ice or water) are included here. The seasonal cycle is plotted for the six lakes (upper panel) and the values of the standard deviation are reported in the middle panel. The number of days with observations is shown in the lower panel.

**Table 2.** Average maximum LSWT and 2m air temperature (indicated as Tair here); lakes are listed in order of decreasing maximum LSWT.

| Lake identifier | D (blue) | B (cyan) | C (green) | A (red) | F (purple) | E (pink) |
|---|---|---|---|---|---|---|
| **LSWT max (°C)** | 13.23 | 13.11 | 11.37 | 9.75 | 7.49 | 6.50 |
| **DoY LSWT max** | 30/07 | 07/08 | 07/08 | 10/08 | 15/08 | 14/08 |
| **Tair max (°C)** | 11.69 | 9.76 | 11.33 | 8.93 | 5.80 | 5.68 |
| **DoY Tair max** | 15/07 | 14/07 | 14/07 | 24/07 | 16/07 | 15/07 |

similar temperatures on average after the end of August, during the cooling period. Lake D and B are the warmest lakes with a very similar maximum average LSWT 13.2±1.6°C and 13.1±1.6°C on the 30th of July and the 7th of August respectively, despite being lakes with different characteristics. As reported in Table 2 the four lakes that are not directly connected to the GrIS peak on average above 10°C. Similar summer maxima have been reported for smaller lakes, not connected to the GrIS, in previous studies (Anderson et al., 2001; Saros et al., 2016).



### 3.3.3 LSWT in relation with Tair and insolation seasonal cycle

Tair and insolation are two very important drivers of LSWT (Woolway et al., 2017). Despite being the coolest lakes among the six, lake E and F receive the highest insolation (Figure 7(b)) being the lakes at the lowest latitude but Tair is the lowest, being the lakes at the highest elevation showing that very likely the insolation will result in a lower temperature (on average) contributing to the melt of the GrIS which in turn generates a cold water influx into the lake which in turn masks the contribution of the insolation. Lake D and B experience very similar maximum temperatures. Lake D is likely shallower than lake B. Lake D is

situated at the lowest latitude among A, B, C and D, and (Figure 7(a)) has the warmest seasonal cycle of air temperature (Tair). Moreover, ERA5-Land indicates higher insolation for lake D than lake B (Figure 7(b)). Based on depth, Tair and insolation, we would expect a higher maximum temperature for lake D compared to lake B. The fact that the observed difference is insignificant raises the speculation that the river inflow of lake D, fed by a colder proglacial lake, is a significant negative heat flux and that this damps the maximum LSWT.

Figure 6(a) suggests that warmer lakes typically experience an earlier timing of maximum temperature while the correspondent Tair and insolation maximums happen at the similar timing. Table 2 reports the maximum annual LSWT and Tair and the days of the year of these for all lake centres. The maximum Tair occurs, on average, in mid-July, except for Tair near lake A which occurs about ten days later. Lake A is located at the highest latitude and is closest to the open sea. The maximum LSWT is higher on average and occurs two to four weeks later than the maximum value of Tair. The longest lag between Tair and

LSWT are calculated for the coldest lakes, E and F, which are connected to the GrIS.

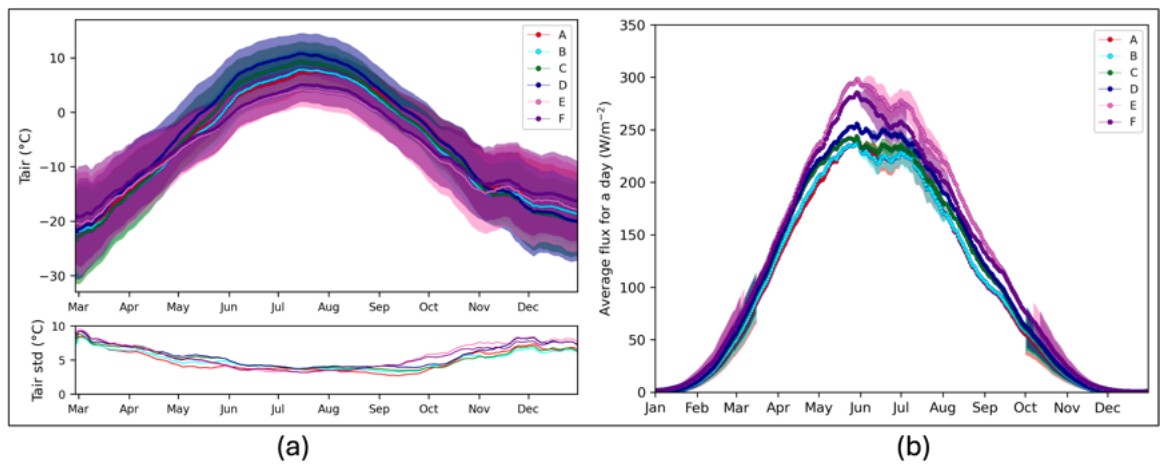

**Figure 7.** Seasonal cycle (daily means from 1995 to 2020 with a 7-day smoothing window) of air temperature (°C) (a) and solar downwelling flux (Wm$^{-2}$) from ERA5-Land reanalysis (b). The seasonal cycle is plotted for the six lakes (upper panel) and for (a) in the lower panel the values of the standard deviation are reported.





### 3.3.4 Asymmetry of the LSWT seasonal cycle

Figure 6(a)(upper plot) shows that all six lakes warm faster and cool more slowly. In contrast, the forcing cycles of Tair and insolation are relatively symmetrical. The asymmetry in LSWT over the summer season is due to the effect of ice cover. Upon ice break up in May, cooler LSWTs are in relatively large disequilibrium with the prevailing Tair and insolation, and the LSWT

rapidly increases in response. During the cooling phase, in contrast, the lake has a steadier lagged response to the progressive decrease in Tair and insolation through late summer and autumn. On average, the earliest retrieved LSWTs at the lake centre are a few degrees above 0°C. This is most likely a sampling effect arising because the first cloud-free observation of the lake centre after ice melt is not immediately after the open water first forms. However, we note that warmer-than-freezing temperatures at the time of ice melt has been reported for lakes in the Tibetan plateau (Kirillin et al., 2021) and Ontario in Canada (Williams,

1969). In the circumstances of dry, cold atmospheric conditions, solar radiation may be relatively strong and the ice snow free, enabling a sufficient flux of insolation through the ice to increase the temperature of the water below, even above 4°C such that there is convective mixing before ice melt. It is uncertain from the satellite observations alone whether such effects are relevant to these lakes.

### 3.4 Temporal variability in lake-mean LSWT

The long term LSWT variations for the six lakes are shown in Figure 8 for the months of June, July, August and September when the lakes are (mostly) ice-free. The warmest months are July and August, when also the temperatures among the lakes differ most. The ranking with respect to temperature tends to be maintained throughout the seasonal cycle.

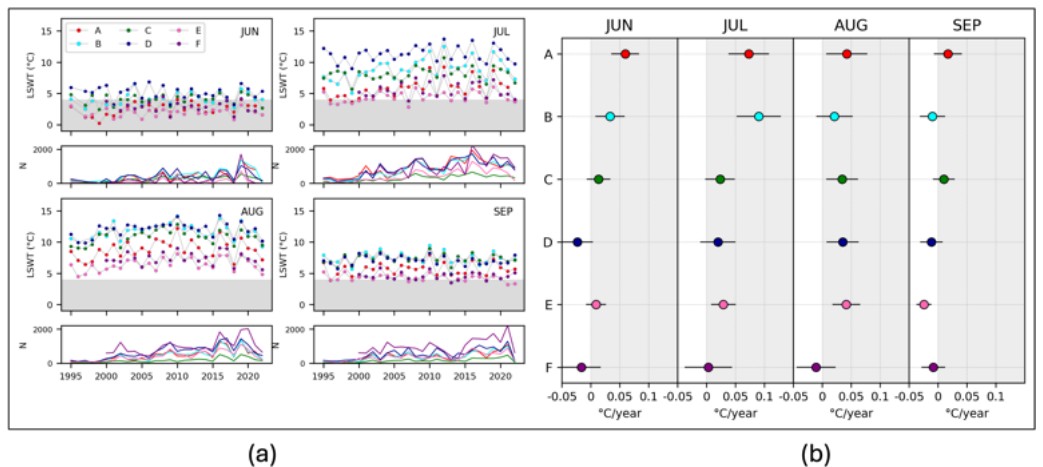

**Figure 8.** (a) Monthly spatial lake mean time series for the months of June, July, August and September for the six lakes (upper plot for each of the months) and the number of observations to create the means (lower plot for each of the months). The grey shaded area features temperatures below 4°C. (b) Trends (linear regression slopes) with standard uncertainty of trend (black bar) of the monthly spatial lake mean time series across years for each month and for each lake. The grey shaded area highlights positive values of the trends.



**Table 3.** Trend comparison between lake A and the closest sea for June, July, August and September.

|  | Jun | Jul | Aug | Sep |
|---|---|---|---|---|
| correlation | 0.44 | 0.62 | 0.69 | 0.72 |
| p-value | 0.02 | 0.0004 | $10^{-5}$ | $10^{-5}$ |
| trend lake A (°C year$^{-1}$) | 0.06±0.02 | 0.07±0.03 | 0.04±0.03 | 0.02±0.02 |
| trend sea (°C year$^{-1}$) | 0.00±0.01 | 0.00±0.02 | 0.03±0.02 | 0.02±0.01 |

In June and July, monthly lake-mean LSWT in lake D is noticeably higher than the other lakes across the years, while in August and September the LSWT of the lakes D, B, C are comparable. Lakes E and F, being connected to the GrIS, do not
exhibit a considerable difference in lake mean temperature throughout the years for any of the months. The six time series of monthly lake-mean LSWTs in Figure 8(a) reveal temporal trends whose values are displayed in Figure 8(b). Twelve of the 15 statistically significant trends in Figure 8(b) are positive (warming over time). The trends for lake A and B in July are respectively 0.09±0.03°C year$^{-1}$ and 0.09±0.04°C year$^{-1}$. Comparing the six lakes, lake A has consistently the highest trends (lake warming at highest rate) across the months. The highest values for the trend of lake A are in July 0.06±0.02°C
year$^{-1}$ and August 0.07±0.03°C year$^{-1}$. This can be related to the fact that lake A is not connected to the GrIS, it is close to the coast and therefore possibly more exposed to meteorological drivers than the other lakes.

To consider the marine influence on lake A, which is close to the coast, SSTs for June to September has been computed from the ESA CCI dataset (Merchant et al., 2019), covering a 2 degree box centred on lake A's latitude immediately to the west. LSWT is 2°C warmer than the nearby SST during July and August, and warmer by less a 1°C in June and September.
The inter-annual correlation between LSWT and SST is positive and statistically significant for all four months. The mean trend across the four months is 0.01±0.01°C year$^{-1}$ in the offshore SST, and is 0.05±0.01°C year$^{-1}$ in the LSWT. Together, the statistics suggest a coupling to the marine climate and an amplified response. The monthly statistics are shown in Table 3. The lakes connected to the GrIS (D, E and F) tend to show smaller interannual trends than the unconnected lakes, although the distinction is not so clear in August. Lake E is overall warming faster than lake F. Lake D experiences contrasting trends,
negative in June and September and positive in July and August. Speculatively, the overall distinction between lakes A, B, and C compared to D, E and F may reflect a negative feedback for the connected lakes via the flux of meltwater. However, lake D is connected only via a river, and, speculatively, warming of the meltwaters during their passage to lake D could act in opposition to this negative feedback, particularly in July and August when the landscape is warmest.

## 3.5 Lake stratification phenology

We estimate lake stratification phenology as the day of the year when the LSWT crosses 3.98°C. On average, warmer lakes in the study region experience an earlier onset and later breakup of thermal stratification. In turn, the cooler lakes, which are directly connected to the GrIS, experience the latest onset and earliest breakup of stratification (Figure 9(c)). Lake F on average reaches ~4°C on the 1st of July and lake E on 7th of July, while for the other lakes this happens between the 11th and 28th





of June. Following the warm season, LSWTs in lake F cools, on average, below ∼4°C on 22nd of September and lake E on
18th of September, while for the other lakes this happens between the 1st and the 12th of October. The time series of the days
of the year of onset and breakup of stratification follow an asymmetric cycle: the break up date is relatively stable across the
six lakes, while the onset tends to occur substantially earlier in the year during the study period, other than for lake F (Figure
9(a)). The largest rates of change in stratification onset are -0.5±0.2 days year$^{-1}$ for lake A and B (Figure 9(b)). The breakup
of stratification only occurs later in the year for lake A (0.2±0.2 days year$^{-1}$ ). The warm stratified season is lengthening for
the three lakes that are not connected to the GrIS (Figure 9), while this is less clear for the connected lakes given the trend
uncertainties.

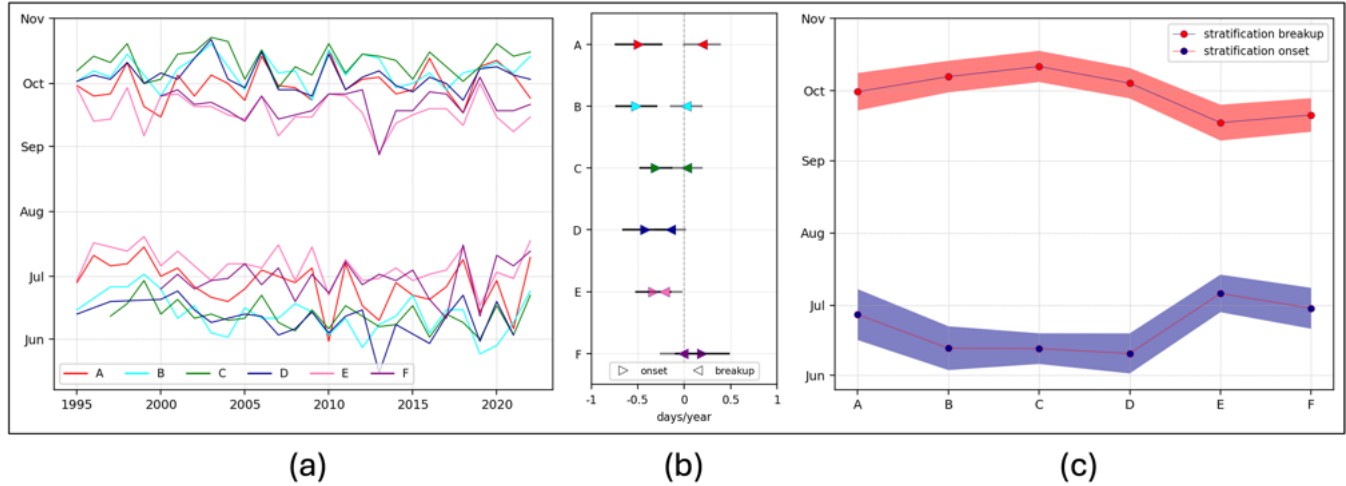

**Figure 9.** (a) Time series of the stratification onset and breakup timing (day of the year) for the six lakes. (b) Trends (linear regression
slopes) with standard uncertainty (black bar) of the stratification onset and breakup timing (days per year) for the six lakes. (c) Average day
of the year for the stratification onset and breakup, where the shaded area represents one standard deviation from the mean, expressing the
variability of the thermal stratification phenology across the years.

### 3.6  Factors influencing the variability in LSWT and stratification phenology

Tair is typically considered the most important atmospheric driver of LSWT (Edinger et al., 1968), but their relationships differ
across seasons. Here, we calculated the correlation (Spearman) between Tair and LSWT anomalies across the studied lakes
(Figure 10) in time. Figure 10(a) and 10(b) show the interannual correlation between the June and August (respectively) LSWT
anomaly and the Tair anomalies of each of the months of the year. For all six lakes, LSWT anomalies in June are best correlated
with Tair anomalies in May (Figure 10(a)). This seems to arise because the lake ice-off date strongly influences the June LSWT
anomaly. Earlier ice-off results in a rapid initial rise of LSWT during summer, thus influencing the June LSWT. Warmer Tair
in May can lead to earlier ice-off. This interpretation is supported by the observation that the stratification onset date is better
correlated with the May Tair anomaly (mean of correlation coefficient across the six lakes: 0.7±0.1) than with that of June





(0.5±0.1), despite stratification onset usually occurring during June. In contrast, LSWT anomalies in August (Figure 10(b)) are most significantly correlated with the August Tair anomalies. During this time the lakes are likely thermally stratified and the volume of water interacting with the atmosphere is relatively small. The exceptions to the high correlation for August are the lakes E and F that directly receive GrIS meltwater. Since warm Tair is expected to increase meltwater flows (but not, for an 370 ice-marginal lake, meltwater temperature), it appears that LSWT-Tair correlation for August is in these cases suppressed by a negative feedback via enhanced cooling by water influx from the GrIS.

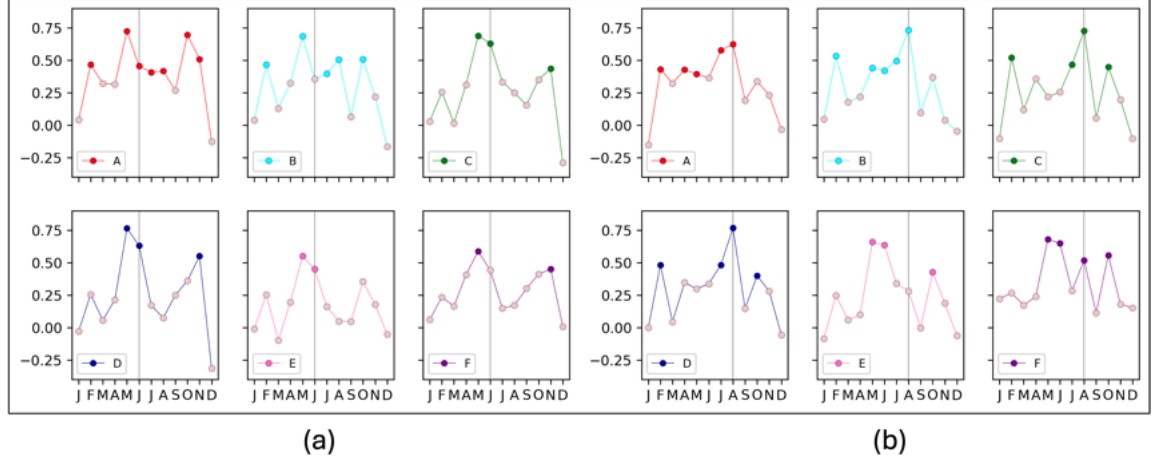

**Figure 10.** Correlations (Spearman) between LSWT anomalies for the month of June (a) and August (b) and the 2m air temperature anomalies from ERA5-Land for all the months in the year. The grey circles indicate a p-value greater than 0.05.

The most important driver of the stratification phenology is air temperature (Woolway et al., 2021). The correlation between stratification phenology and Tair is found to be relevant and significant only for the onset, while for the break up only mild correlations appear for the lakes connected to the GrIS and lake A. For lake D, which is connected to the GrIS through a long 375 stream, the correlations with Tair are not significant, indicating that the changes in the stratification timing are driven by other factors. The highest correlations are found for stratification onset for the lakes directly connected to the GrIS (lake E and F) and lake A. Correlations with the shortwave solar radiation anomalies have been investigated using the surface solar irradiance values from Era5-Land and using the ESA Climate Change Initiative Cloud observations (Stengel et al., 2020). No significant correlation has been found for any of the lakes with either dataset.

**4 Discussion**

Lakes are important features of West Greenland's seasonally ice-free landscape, their biogeophysical interactions in summer being in part determined by their energy budget and the LSWT they attain. In this study, we exploited quantitative thermal remote sensing to present new climatological LSWT information for lakes across this landscape. Analyzing data from 138 lakes



observable in Landsat 8 imagery, we observed a wide spectrum of maximum LSWTs varying from 5°C to 20°C, alongside
varying surface albedo values spanning 0.03 and 0.14. Notably, high-albedo lakes, often connected directly to the GrIS or indi-
rectly through river systems, tend to exhibit cooler temperatures. This cooling effect is attributed to their enhanced reflectivity,
which reduces solar absorption, coupled with the influence of cold glacial meltwater inflows. Conversely, smaller and typically
shallower lakes with lower albedo tend to absorb more solar radiation and consequently reach warmer temperatures (Read and
Rose, 2013; Heiskanen et al., 2015). These findings underscore the intricate interplay between lake morphology, solar radiation
dynamics, and climatic influences, essential for understanding the response of these ecosystems to climate change.

In addition to the broader analysis across the region, our study also investigated the specific dynamics of six lakes in West
Greenland. These lakes were selected based on their contrasting elevations and proximity to the GrIS, allowing us to examine
their seasonal climatology and interannual variability in detail. Meteorological sensors provided critical data, capturing a time
series of LSWT and LIC observations that complement the temporal coverage limitations of Landsat imagery. Despite their
capability to resolve only the largest lakes, these sensors enabled us to uncover nuanced insights into the thermal dynamics of
these water bodies. These six studied lakes had contrasting elevation and proximity to the GrIS, as well as to the sea. Seasonally,
these lakes exhibit significant variability in ice cover, typically remaining ice-free from June through to November, although
the exact timing of complete ice reformation is affected by some observational uncertainty. This seasonal pattern, notably
asynchronous with the solar insolation cycle, underscores the complex interplay of climatic factors influencing lake dynamics.
Air temperature conditions leading up to the ice-free season, particularly in May, emerge as a pivotal factor influencing both
stratification onset and the timing of ice-off events. Warmer temperatures during this period accelerate the melting of lake ice,
influencing stratification dynamics and subsequent thermal regimes. The observed interannual variability in these processes
highlights the sensitivity of West Greenland lakes to fluctuations in atmospheric conditions, emphasizing their role as indicators
of broader climatic trends in the region.

The detailed examination of these lakes reveals distinct thermal dynamics and responses to environmental stimuli. Lakes
not directly connected to the GrIS typically reach peak temperatures ranging from 9°C to 14°C during the summer months,
exceeding average air temperatures due to direct solar heating. In contrast, ice-marginal lakes exhibit cooler peak temperatures
around 7°C, a phenomenon not fully explained by their elevated albedo alone. The influx of cold meltwater from nearby
glaciers likely acts as a significant cooling mechanism, influencing the thermal balance and seasonal variability of these water
bodies. The high albedo of ice-marginal lakes suggests substantial influx of mass and meltwater from the GrIS. This meltwater
introduces a negative heat flux that correlates with August air temperatures, affecting glacial melting dynamics. The lack
of correlation arises from competing mechanisms, notably warming from solar radiation and cooling from meltwater influx
associated with warmer summer air temperatures, complicating the LSWT response to climate change.

We found that the warm stratified season of the studied lakes typically spans from mid-June to early October, though
slightly shorter for cooler ice-marginal lakes. The onset of stratification correlates closely with air temperature in the preceding
month, influencing the rate of lake ice melt. Our study found no significant influence of insolation on this process. Over the
months of June to September, LSWT trends varied from insignificantly negative to warming trends of 0.5°C to 1°C decade$^{-1}$
across different lakes. Stratification onset appears to be advancing by approximately 5 to 8 days per decade, while the end



of stratification shows more stability over time, consistent with studies from other regions worldwide (Woolway et al., 2021).
These findings suggest that ice-marginal lakes may increasingly interact with the dynamics of the GrIS margin, potentially becoming more important in the future (Mallalieu et al., 2021; Carrivick et al., 2022). Changes in water temperature and stratification phenology substantially impact light availability, nutrient cycling, and oxygen levels crucial for lake ecosystems (Woolway et al., 2021). Surface water temperature serves as a 'sentinel' of climate change, integrating various climatic and in-lake drivers that influence the lake's surface energy budget (Woolway et al., 2021; Schneider and Hook, 2010). Understanding
these dynamics is crucial for assessing the resilience of West Greenland's lakes to ongoing environmental changes and their implications for ecosystem health and function.

Observations of the temperature of pro-glacial lakes in this landscape are sparse. It has been assumed in some mass balance models of glaciers terminating in lakes that the lakes don't warm to more than 1°C (Truffer and Motyka, 2016; Chernos et al., 2016), a notion contradicted by the remotely sensed data analyzed in this study. This discrepancy underscores the necessity
of leveraging remote sensing to complement sparse in situ data in regions like West Greenland. However, only large lakes (of area at least 3km$^2$) can be observed with good temporal resolution by meteorological satellites (used for ESA CCI Lakes dataset) which offer a sufficiently long time series to detect climatic signals and well-characterised uncertainty and stability. For smaller lakes, higher spatial resolution satellites, such as Landsat 8, can offer observations at a finer spatial resolution (100m) but a much longer revisiting time is often insufficient for a good description of the temporal variations. These are
current limitations of the remote sensing data. Furthermore, lake glacier interaction mechanism could be better investigated through new field work. Collecting ground truth data would be very useful to more precisely quantify the interactions between lakes and the ice sheet and to unravel the complex thermal structure at various depths of these lakes and how it varies in space and time. Such approaches are essential for elucidating the complex relationships between climatic variables, ice dynamics, and lake behaviors, providing critical insights into ecosystem responses to climate change in such a key region for local and
global climate.

*Data availability.* All the datasets utilized for the current study are all available online as open data and in particular: the digital elevation model ASTER GDEM Version 3 (ASTGTM) is available at the Land Processes Distributed Active Archive Center (LP DAAC) https://doi.org/10.5067/ASTER/ASTGTM.003, the bathymetry GLOBathy dataset is available at figshare https://doi.org/10.6084/m9.figshare.c.5243309.v1, the ESA CCI Lakes - LWST and LIC dataset available at the Centre for Environmental Data (CEDA) archive http://dx.doi.
org/10.5285/a07deacaffb8453e93d57ee214676304, the ESA CCI Clouds dataset is available at the Deutscher Wetterdienst archive https://dx.doi.org/doi:10.5676/DWD/ESA_Cloud_cci/AVHRR-AM/V003, the ESA CCI SST dataset is available at the CEDA archive http://dx.doi.org/10.5285/62c0f97b1eac4e0197a674870afe1ee6, and the ECMWF ERA 5 Land data are available at the Copernicus Climate Data Store https://doi.org/10.24381/cds.e2161bac. The Landsat 8 data are available at the Landsat Data Access web page https://www.usgs.gov/landsat-missions/landsat-data-access where to discover how to search and download all Landsat products from United States Geological
Survey (USGS) data portals. The USGS Landsat no-cost open access data policy remains intact since its inception in 2008. Also, all the datasets used and/or analysed during the current study are available from the corresponding author on reasonable request.



*Author contributions.* L.C. and C.M. designed the study, performed the analysis, interpreted the results, and wrote the manuscript. R.I.W. contributed in writing and revising the manuscript. N.M. dealt with the extraction of the Landsat 8 data.

*Competing interests.* The authors declare no competing interests.

*Acknowledgements.* The authors acknowledge the European Space Agency for this work with gratitude. The European Space Agency supported the Climate Change Initiative - New ECVs for Lakes, which in turn has has provided the majority of support leading to the outcomes herein described, via grant reference 4000125030/18/I-NB. The authors acknowledge the United Kingdom National Center for Earth Observation (NCEO) for funding the development of some of the software used to process Landsat 8 data, courtesy of the U.S. Geological Survey and also acknowledge the United Kingdom National Environment Research Council (NERC) for funding part of this work within the project

on Greenland ice marginal lake evolution as driver of ice sheet change (NERC reference: NE/X013537/1).



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
