# Peer review of "Factors influencing lake surface water temperature variability in West Greenland and the role of the ice sheet"

_EGUsphere, 2024_

## Author Response (AR1)

This manuscript presents findings from six lakes in Southwest Greenland through the compilation and analysis of a plethora of remote sensing datasets. The results provide a thorough and interesting insight into the seasonal cycle of lake surface water temperature (LSWT) and ice cover (LIC), with an assessment of meltwater fed and non-meltwater fed lakes, maritime influence, and meteorological drivers. The manuscript concludes that the interannual variability in LSWT is predominantly driven by air temperature, and highlights the importance of continuing long-term remote sensing efforts and in situ monitoring to further our understanding of lake dynamics in Greenland.

In all, the manuscript effectively collates multiple datasets to investigate lakes in Greenland at an impressive level of detail. I think it also nicely demonstrates how many remote sensing datasets are readily available for constructing valuable and insightful analysis, in regions where remote sensing analysis remains in its infancy. My main feedback to the authors is on how the findings are presented through the manuscript's structure. The Results section contains interpretation of complex datasets alongside the presentation of the results, making it a very long read that I, as a reader, got lost in many times. I also think this structuring inhibits discussion of the findings, with support/comparison from the work of others from Greenland and the wider Arctic. I have made a suggestion of an alternative structure to better convey the results, interpretation, discussion and conclusions to the reader. With re-structuring, along with my other comments (detailed below), I think this manuscript would be a valuable and interesting contribution to The Cryosphere. Thank you for a very enjoyable read!

**RESPONSE:** *The authors thank the reviewer for the suggestions to improve the readability of this complex paper especially in relation to the reorganisation of the results and the discussion sections. The authors thank the reviewer for pointing to the Greenland lake databases.*

**Main comments**
**1. The manuscript structure**
In its current form, the manuscript's Results section includes interpretation alongside data presentation, which makes it quite extensive and occasionally difficult to navigate. This structure may also reduce the opportunity to provide a more in-depth discussion, as some interpretations are dispersed throughout the Results and Discussion. I recommend restructuring the manuscript to present interpretation in a dedicated section, which would allow for a clearer and more focused discussion supported by relevant evidence from other studies.
The following structure could help improve clarity, with the Results section focusing exclusively on the presented figures and findings, and the Discussion section providing interpretation, supporting references, and comparisons with other studies:
3. Results
3.1 Physical characterization of the study region
3.2 Characterization of the six studied lakes
3.3 LSWT and LIC
3.4 Lake stratification phenology

4. Discussion
4.1 Seasonal trends in LSWT and LIC
4.2 Influence of air temperature and insolation on LSWT and lake stratification
4.3 Temporal variability in LSWT
4.4 Implications for LSWT and LIC studies in Greenland and the Arctic
5. Conclusions

Most of the proposed sections already exist in the manuscript; however, additional discussion that compares findings with other studies would be helpful, particularly in Section 4.3 and 4.4. As the authors note, there are few existing studies on Greenlandic lakes, but incorporating insights from studies on lakes in the broader Arctic region would enhance this section.

*RESPONSE: Thank you for your suggestions. We agree that these changes improve the readability of the paper and have altered the text accordingly.*

*4. Discussion*

*In this study, we exploited quantitative high-resolution thermal remote sensing to present new climatological LSWT information for lakes across this landscape. In addition to the broader analysis across the region, our study also investigated the specific dynamics of six lakes in West Greenland using meteorological sensors which complement the temporal coverage limitations of Landsat imagery capturing a time series of LSWT and LIC observations. The six lakes were selected based on their contrasting elevations and proximity to the GrIS, allowing us to examine their seasonal climatology, interannual variability and the influence of atmospheric variables on the lake dynamics in detail.*

*4.1 Spatial thermal characteristics of lakes in the region with high resolution sensor*
*High-resolution thermal remote sensing allowed to characterise the lakes in the region spatially only since currently high-resolution sensors have low revisiting time. Analysing data from 138 lakes observable in Landsat 8 imagery, we observed a wide spectrum of maximum LSWTs varying from 5°C to 20°C, alongside varying surface albedo values spanning 0.03 and 0.14. Notably, high-albedo lakes, often connected directly to the GrIS or indirectly through river systems, tend to exhibit cooler temperatures. This cooling effect is attributed to their enhanced reflectivity, which reduces solar absorption, coupled with the influence of cold glacial meltwater inflows. Conversely, smaller and typically shallower lakes with lower albedo tend to absorb more solar radiation and consequently reach warmer temperatures (Read et al., 2013, Heiskanen et al., 2015). These findings underscore the intricate interplay between lake morphology, solar radiation dynamics, and climatic influences, essential for understanding the response of these ecosystems to climate change.*

*4.2 Seasonal trends in LSWT and LIC*
*Meteorological sensors provided critical time series of LSWT and LIC observations that enabled us to uncover nuanced insights into the thermal dynamics of these water bodies. These six studied lakes had contrasting elevation and proximity to the GrIS, as well as to the sea. Seasonally, these lakes exhibit significant variability in ice cover,*

typically remaining ice-free from June through to November, although the exact timing of complete ice reformation is affected by some observational uncertainty. This seasonal pattern, notably asynchronous with the solar insolation cycle, underscores the complex interplay of climatic factors influencing lake dynamics.

The detailed examination of these lakes reveals distinct thermal dynamics and responses to environmental stimuli. Lakes not directly connected to the GrIS typically reach peak temperatures ranging from 9°C to 14°C during the summer months, exceeding average air temperatures due to direct solar heating. In contrast, ice-marginal lakes exhibit cooler peak temperatures around 7°C, a phenomenon not fully explained by their elevated albedo alone. The influx of cold meltwater from nearby glaciers likely plays an important role impacting the thermal balance and seasonal variations of these water bodies.

Analysing the seasonal cycle, we found that the warm stratified season of the studied lakes typically spans from mid-June to early October, though slightly shorter for cooler ice-marginal lakes and that all six lakes warm faster and cool more slowly due to the effect of the ice cover. In contrast, the forcing cycles of Tair and insolation are relatively symmetrical. Upon ice break up in May, cooler LSWTs are in relatively large disequilibrium with the prevailing Tair and insolation, and the LSWT rapidly increases in response. During the cooling phase, in contrast, the lake has a steadier lagged response to the progressive decrease in Tair and insolation through late summer and autumn.

**4.3 Temporal variability in LSWT**

The observed interannual variability in these processes highlights the sensitivity of West Greenland lakes to fluctuations in atmospheric conditions, emphasizing their role as indicator of broader climatic trends in the region.

Over the months of June to September, LSWT trends varied from insignificantly negative to warming trends of 0.5°C to 1°C decade$^{-1}$ across different lakes with clear differences for ice-marginal lakes, coastal lakes and inland lakes. Stratification onset appears to be advancing by approximately 5 to 8 days per decade, while the end of stratification shows more stability over time, consistent with studies from other regions worldwide (Woolway et al., 2021).

Our findings suggest that ice-marginal lakes may increasingly interact with the dynamics of the GrIS margin, potentially becoming more important in the future (Mallalieu et al., 2021; Carrivick et al., 2022). Changes in water temperature and stratification phenology substantially impact light availability, nutrient cycling, and oxygen levels crucial for lake ecosystems (Woolway et al., 2021). Surface water temperature serves as a 'sentinel' of climate change, integrating various climatic and inlake drivers that influence the lake's surface energy budget (Woolway et al., 2021; Schneider and Hook, 2010). Understanding these dynamics is crucial for assessing the resilience of West Greenland's lakes to ongoing environmental changes and their implications for ecosystem health and function.

**4.4 Influence of air temperature and insolation on LSWT and lake stratification**

*Air temperature conditions leading up to the ice-free season, particularly in May, emerge as a pivotal factor influencing both stratification onset and the timing of ice-off events. The onset of stratification correlates closely with air temperature in the preceding month, influencing the rate of lake ice melt. Our study found no significant influence of insolation on this process.*

*Warmer temperatures during the period leading to the ice-free season accelerate the melting of lake ice, influencing stratification dynamics and subsequent thermal regimes. The influx of cold meltwater from nearby glaciers likely acts as a significant cooling mechanism, influencing the thermal balance and seasonal variability of these water bodies. The high albedo of ice-marginal lakes suggests substantial influx of mass and meltwater from the GrIS. This meltwater introduces a negative heat flux that correlates with August air temperatures, affecting glacial melting dynamics. The lack of correlation with August air temperature for ice-marginal lakes arises from competing mechanisms, notably warming from solar radiation and cooling from meltwater influx associated with warmer summer air temperatures, complicating the LSWT response to climate change.*

*4.5 Implications for LSWT and LIC studies in Greenland and the Arctic*

*Observations of the temperature of pro-glacial lakes in this landscape are sparse. It has been assumed in some mass balance models of glaciers terminating in lakes that the lakes do not warm to more than 1°C (Truffer and Motyka, 2016; Chernos et al., 2016), a notion contradicted by the remotely sensed data analysed in this study. This discrepancy underscores the necessity of leveraging remote sensing to complement sparse in situ data in regions like West Greenland. However, only large lakes (of area at least $3km^2$) can be observed with good temporal resolution by meteorological satellites (used for ESA CCI Lakes dataset) which offer a sufficiently long time series to detect climatic signals and well-characterised uncertainty and stability.*

*For smaller lakes, higher spatial resolution satellites, such as Landsat 8, can offer observations at a finer spatial resolution (100m) but a much longer revisiting time is often insufficient for a good description of the temporal variations. An example is the use of the high spatial resolution sensor ASTER to characterise the thermal behaviour of lakes in Arctic Sweden where the measurements considered were very sparse in time (Dye et al.,2021). These are current limitations of remote sensing data, which will be partly addressed with improvements in spatial and temporal resolution such as those provided by the satellites for thermal remote sensing such as TRISHNA (Lagouarde et al., 2018) to be launched after 2025. Furthermore, lake-glacier interaction mechanism could be better investigated through new field work. Collecting ground truth data would be very useful to more precisely quantify the interactions between lakes and the ice-sheet and to unravel the complex thermal structure at various depths of these lakes and how it varies in space and time. Such approaches are essential for elucidating the complex relationships between climatic variables, ice dynamics, and lake behaviours, providing critical insights into ecosystem responses to climate change in such a key region for local and global climate.*

*5. Conclusions*

*Lakes are important features of West Greenland's seasonally ice-free landscape, their biogeophysical interactions in summer being in part determined by their energy budget and the LSWT they attain. In this study, we presented new long-term consistent LSWT information for lakes across this landscape using satellite data. A broader analysis across the region to characterise thermally the lakes in space was achieved with high spatial resolution data from Landsat-8. Investigations on specific dynamics were conducted on six lakes in West Greenland using high temporal resolution meteorological sensors. The six lakes are located at contrasting elevations and proximity to the GrIS and to the coast, allowing us to unveil their long term thermal behaviour (seasonal climatology and interannual variability) for the first time and to attempt to disentangle the intricate interplay between lake morphology, solar radiation dynamics, and climatic influences the influence of atmospheric variables on the lake dynamics in details and in a consistent manner.  Contrary to  some mass balance models of glaciers terminating in lakes, we found that these lakes warm to much more than 1°C reaching on average a water temperature between 6.6 and 7.5°C. The lakes not connected to the ice-sheet reach an average surface water temperature of more than 13°C. The most important atmospheric driver is air temperature. However, the ice-sheet significantly impacts the thermal behaviour of the lakes, frequently overshadowing the atmospheric signatures. For example, no correlation is found in summer between LSWT anomalies and Tair only for the ice marginal lakes, indicating a mitigation of Tair impact on the lakes by an enhanced cooling of water influx from the GrIS.*
*The vicinity of the coast also influences lake thermal state. The lake temperature is influenced by the marine climate and a comparison with nearby SST shows that the lake amplifies the response, exhibiting consistently the largest trend in temperatures and in stratification onset and breakup timing among the six lakes.*

**2. Greenlandic names for the lakes**

I noticed that the Greenlandic names for the lakes studied are not used in this manuscript, despite these names being well documented and available through the Language Secretariat of Greenland (Oqaasileriffik, https://oqaasileriffik.gl) and the QGreenland dataset (Moon et al., 2023, https://qgreenland.org/).

According to the placename database, the lakes presented here have the following names (in New Greenlandic):

Lake A – Eqalussuit Tasiat
Lake B – Nassuttuutaata Tasia
Lake C – Itinnerup Tasersua
Lake D – Tasersuaq Aallaartagaq (D)
Lake E – Ammalortoq (E)   (please verify, based on lake extent in Figure 1)
Lake F – Tarsartuup Tasersua (F)

I suggest revising the manuscript to incorporate these Greenlandic placenames rather than the A-F convention. Using the proper Greenlandic names would be a meaningful step toward ensuring accurate place naming and is particularly important given that Greenlandic placenames are sometimes underrepresented or misapplied in Cryosphere/Greenland-based research. Adopting these names would strengthen the manuscript's alignment with locally recognised standards and best practices in geographic terminology.

*RESPONSE: Thank you for pointing out to us the databases. This is very useful! We have kept the A-F conventions for the figures since the names are very long but reported names and convention throughout the paper.*

**3. Improved figure caption descriptions**

In several instances, figure captions are somewhat brief, with much of the detail given in the main text. Adding more information directly in the figure captions, such as data sources and general descriptions, would allow readers to more easily interpret each figure independently. This applies particularly to Figures 1, 2, 4, 5, and 7, and I have included specific suggestions in the minor comments below.

*RESPONSE: **Thank you.** We have expanded the text in the captions to make clearer the content of the figures.*

**Minor comments**

L19-21: I wouldn't say that this interaction is between LSWT and ice margin dynamics directly, as there are many other lake processes that influence ice margin dynamics. I would suggest changing this to encompass all lacustrine lake-ice processes, with references to work where processes have been studied in lacustrine settings; for instance calving (e.g. Mallalieu et al., 2021; Minowa et al., 2023), submarine melting (e.g. Sugiyama et al., 2021), lake temperature (e.g. Dye et al., 2021), and GLOFs (e.g. Kjeldsen et al., 2017; Grinsted et al., 2017).

*RESPONSE: We agree. We have changed the following sentence:*

***Specifically, the interaction between LSWT and ice margin dynamics can accelerate glacier mass loss, particularly in areas where lakes abut ice fronts.***

*with:*

***Specifically, the interaction between lakes and ice margin involving lake-ice processes such as instance calving (Mallalieu et al. 2021, Minowa et al.,2023), submarine melting (Sugiyama at al., 2021), lake temperature (Dye et al.,2021), and glacial lake outburst floods (GLOFs) (Kjeldsen at al., 2017, Grinsted et al., 2017) can accelerate glacier mass loss, particularly in areas where lakes abut ice fronts.***

L25: How sparse are in situ LSWT measurements in Greenland? Could you provide an overview of the few in situ studies available?

*RESPONSE: We have found few studies reporting temperature measurements on lakes:*

- *[Saros et al. 2016, Thermal stratification in small arctic lakes of southwest Greenland affected by water transparency and epilimnetic temperatures, https://doi.org/10.1002/lno.10314] The paper reports temperature measurements taken in 2013 on 22 lakes  and then in 2014 taken again on a subset of 8 lakes in southwest Greenland  around Kangerlussuaq. All the lakes*

*are very similar being quite small (surface area between 0.022 and 0.824 km2) and moderately deep (9–36 m). None of them are connected to the ice sheet.*

- *[Hazukova et al 2024, Earlier ice melt increases hypolimnetic oxygen despite regional warming in small Arctic lakes, https://doi.org/10.1002/lol2.10386] The paper presents temperature measurements taken between 2011 and 2022 in west Greenland on 13 lakes (mainly) between the ice sheet and Kellyville in Kangerlussuaq. All the lakes are very similar being quite small (surface area between 0.6 and 0.37 km2) and moderately deep (10.9–28.8 m). The 13 lakes are a subset of the 22 of [Saros et al 2016].*
- *[Hazukova et al 2022, Under Ice and Early Summer Phytoplankton Dynamics in Two Arctic Lakes with Differing DOC, https://doi.org/10.1029/2020JG005972] Temperatures for 2 lakes (of the 22 used in [Saros, 2016]) collected in 2019 are used for the study)*
- *[https://doi.org/10.17897/BKTY-J070] In situ monitoring of two lakes in Kobbefjord in south-west Greenland (Badesø / Kangerluarsunnguup Tasia: 64,13ºN, 51,36ºW and Qassi Sø: 64,15ºN, 51,31ºW) at 2m and 10m from 2009 to 2019.*
- *[Kettle et al 2004, Empirical modelling of summer lake surface temperatures in Southwest Greenland, https://doi.org/10.4319/lo.2004.49.1.0271] For ~ 30 lakes across a E–W transect from the ice sheet near Kangerlussuaq to the outer coast south of Sisimiut, temperatures were measured in the period 1998-2000 (although part of the lakes were monitored only in 1998-1999 and the rest only in 1999-2000). Same dataset as [Anderson et al 1999], [Brodersen et al 2000], [Brodersen et al 2002].*
- *[Anderson et al 1999, Limnological and paleolimnological studies of lakes in south-western Greenland, https://doi.org/10.34194/ggub.v183.5207] Lake surface water temperature measured in 1998-2000 on >30 lakes. Same dataset as [Kettle et al 2004], [Brodersen et al 2000], [Brodersen et al 2002].*
- *[Brodersen et al 2000, Subfossil insect remains (Chironomidae) and lake-water temperature inference in the Sisimiut–Kangerlussuaq region, southern West Greenland, https://doi.org/10.34194/ggub.v186.5219] Lake surface water temperature measured in 1998 on 17 lakes, in 1999-2000 on 31 lakes. Same dataset as [Kettle et al 2004], [Anderson et al 1999], [Brodersen et al 2002].*
- *[Brodersen et al 2002, Distribution of chironomids (Diptera) in low arctic West Greenland lakes: trophic conditions, temperature and environmental reconstruction, southern West Greenland, https://doi.org/10.1046/j.1365-2427.2002.00831.x] Lake surface water temperature measured in 1999 on 29 lakes. Same dataset as [Kettle et al 2004], [Anderson et al 1999], [Brodersen et al 2000].*

*We have added the following text with references to these articles:*

**Few studies have been found in literature reporting temperature measurements on lakes in Greenland which have been collected in two main campaigns. Measurements taken between 2011 and 2022 on small similar lakes (from a minimum of 13 lakes and a maximum of 22 of surface area between 0.022 and 0.824 km² ) in southwest Greenland around Kangerlussuaq have been reported in studies on thermal stratification (Saros et al., 2016),**

*under ice phytoplankton dynamics (Hazukova et al., 2021) and on the relation between ice melt timing and dissolved oxygen concentration (Hazukova et al., 2024). None of the lakes was connected to the GrIS. The other campaign produced water temperature measurements on about 30 lakes across a E–W transect from the ice sheet near Kangerlussuaq to the outer coast south of Sisimiut during the period 1998-2000 and the studies reporting the dataset focussed on empirical modelling of summer lake water temperature in Greenland lakes (Kettle et al., 2004) and on limnology and paleolimnology of lakes in Greenland (Anderson et al., 1999; Brodersen and Anderson, 2000, 2002). In addition to the measurements reported in the above studies, the Greenland Ecosystem Monitoring BioBasis programme (GEM, 2025) recorded water temperature from 2009 to 2019 on two small lakes in Kobbefjord, Badesø / Kangerluarsunnguup Tasia and Qassi Sø, at 2m and 10m respectively.*

L26-27: I beg to differ on this point. I think yes, there are logistical costs and monitoring challenges in Greenland. However, there are research institutes and local populations that live relatively close to lakes for monitoring. I would say instead that monitoring is limited to these areas close to settlements, and access to remote regions (i.e. far from a well-connected settlement) remains challenging.

*RESPONSE: Agreed. We have changed the following sentence:*

*Indeed, the logistical costs and monitoring challenges in this remote region, especially over decadal timescales, have hindered comprehensive research efforts.*

*into*

*Indeed, monitoring especially over decadal time scales is limited to areas close to settlements, and access to remote regions (i.e. far from a well-connected settlement) remains challenging.*

L43-44: "We have included in our study lakes that connected to…" >> "We have included in our study lakes that are connected to…"
L53: "(masl)" >> "(m a.s.l.)"
L53: "The edge of the GrIS…" >> "The margin of the GrIS…"

*RESPONSE: Thank you for noticing it! This has now been changed.*

L64-66, 66-68: This overview of climatic conditions, next to the GrIS and towards the coast, is in reference to a relatively old study (Anderson et al., 2001). Please can this be updated reference to newer studies, or if such a reference does not exist, can you summarise the climatic conditions from data sources that are representative of the area. Data sources that I think could be useful are the DMI/Asiaq weather station networks for land observations, and the PROMICE weather stations for near-/on-ice

observations (the KAN station transect in particular). Another option could be to look at updating the summary with the ECWMF ERA-5 land data that is used in this study.

*RESPONSE: We agree. We have updated the information on climatic conditions and the reference.*
*We have changed the following:*

**The climate of the study region also varies considerably, principally from the coast to the area next to the GrIS (Anderson et al., 2001), with consequent changes in vegetation and lake properties. Areas close to the GrIS have a Low Arctic continental climate, with mean annual surface air temperature of -6°C, an annual temperature range of ~30°C, continuous permafrost, and low precipitation << 150 mm year$^{-1}$. The zone from the GrIS to mid-way towards the coast is characterized by very low (negative) effective precipitation. The coast is characterised by a Low Arctic maritime climate, with a lower annual temperature range (~25°C) and higher precipitation (300 mm year$^{-1}$). The moderated summer maximum temperatures and the presence of coastal fog banks associated with the more maritime climate imply a longer presence of snow and snow packs in this region (Anderson et al., 2001).**

*With the following:*

**The climate of the study region also varies considerably, principally from the coast to the area next to the GrIS (Cappelen et al., 2021), with consequent changes in vegetation and lake properties. Areas close to the GrIS have a Low Arctic continental climate, with mean annual surface air temperature of -5°C, an annual temperature range of ~30°C, continuous permafrost, and low precipitation < 170 mm year$^{-1}$. The zone from the GrIS to mid-way towards the coast is characterized by very low (negative) effective precipitation. The coast is characterised by a Low Arctic maritime climate, with a lower annual temperature range (~22°C) and higher precipitation (~500 mm year$^{-1}$). The moderated summer maximum temperatures and the presence of coastal fog banks associated with the more maritime climate imply a longer presence of snow and snow packs in this region (Cappelen et al., 2021).**

L73-74: "glacier terminal lakes" is not a common term. I think a more used and suitable term is "ice marginal lakes" or "proglacial lakes".

*RESPONSE: Thank you. Changed.*

L83-85: Please provide the spatial resolution of the DEM product here.

*RESPONSE: The sentence:*

**..., provides a global digital elevation model (DEM) of land surface on Earth that extends from 83°N to 83°S, has at a horizontal spatial resolution of 1 arc second**

*(about 30 metres at the equator). The data are projected onto the 1984 World Geodetic System (WGS84)/1996 Earth Gravitational Model (EGM96) geoid. …*

*has been changed into*

*… . It provides a global digital elevation model (DEM) of land surface on Earth that extends from 83°N to 83°S. The data are projected onto the 1984 World Geodetic System (WGS84)/1996 Earth Gravitational Model (EGM96) geoid and the spatial resolution is 1 arc second (about 30 metres at the equator) and they have been resampled at 50 metres resolution to be used with Landsat 8 data for this work. The ASTER GDEM  version 3 dataset …*

L89: What is "reasonable accuracy"? Can you provide an error estimate here for this dataset?

*RESPONSE: It is very difficult to estimate an error for the bathymetry. In the paper, a validation is carried out for a small portion of the 1.5 million lakes using various metrics and the authors reports a "reasonable accuracy". The lake bathymetry is derived with a model which has been selected among few candidates. The selection is based on the comparison of the predicted maximum depth with the observed value for about 1500 lakes giving NRMSE = 0.17, and $\rho$ = 0.94 for the selected model. Also, a cross validation has been carried out. The actual validation of the bathymetry reported on the paper consists on a visual comparison of the predicted bathymetry with the observed bathymetry for 8 lakes. Regarding the six lakes of this paper, we have found that the maximum depth of one of them is unrealistic and we have pointed it out in the paper.*

*We have changed and added the following text at line 89:*

*GLOBathy provides estimates of bathymetry for lakes worldwide with reasonable accuracy, given the complexity of estimating underwater topography, as reported in the article. The bathymetry is derived with a model which has been selected among few candidates. The selection is based on the comparison of the predicted maximum depth with the observed value for about 1500 lakes giving  the root mean squared error normalized with standard deviation, NRMSE = 0.17, and the Spearman's Rho correlation coefficient , $\rho$= 0.94 for the selected model. Also, a cross validation has been carried out. The actual validation of the bathymetry reported on the paper consists of a visual comparison of the predicted bathymetry with the observed bathymetry for eight lakes. Regarding the six lakes of this paper, we have found that the maximum depth of Lake Eqalussuit Tasiat is unrealistic (see Section 3.2).*

Figure 1: Where have the satellite mosaic, Greenland outline, and outlines for the lakes been sourced from? I also think panel A can be inset into panel B, rather than having panel A the same size as panel B.

*RESPONSE: The Greenland outline has been sourced from the python basemap module, the lake outline has been created by us  for the ESA CCI LAKES and derived*

*from the GloboLakes mask. We have added some text in the ESA CCI Lakes Section. The following text has been added to the caption of Figure 1:*

 ***The lake mask shown in the plot is from the ESA CCI LAKES dataset (Carrea et al, 2022b) and it is used for this work.***

L103: "respectevely" >> "respectively"

*RESPONSE: Changed.*

L103-106: I understand that the relevant references explain in detail how LSWT is derived from these satellites, but for the reader here, can you provide a summary of the methodology for deriving LSWT.

*RESPONSE: Thank you for this suggestion. We have now added the following text at line 108 to summarise the methodology:*

***The lake mask is available at zenodo (Carrea et al, 2022b) and it was derived from the GloboLakes mask (Carrea et al, 2015). For LSWT, the optimal estimation retrieval method of (MacCallum and Merchant, 2012) was applied on image pixels identified as water according to both the lake mask and a reflectance-based water detection scheme (Carrea et al., 2023) which was specifically designed to distinguish water from non-water pixels such as clouds, ice or land.***

L107-108: I am not sure I fully understand this. Have LSWT and LIC been upsampled from 1 km (approx.) to a different spatial resolution? I see in Figure 5 a demonstration of the number of observations for one of the lakes, but I remain unsure. Also, for Figure 5, are these observations from the ESA CCI Lakes dataset? Clarification in figure caption is needed, along with a scalebar to demonstrate the scale of each pixel in the figure.

*RESPONSE: The grid of the satellite measurement is not regular and therefore difficult to handle when used. The LSWTs have been retrieved in the original grid and then regridded to a regular grid of 1/120° (which is about 1km at the equator) so that the LSWT data are easier to handle. The number of observations reported in Figure 5 are the observations from the ESA CCI Lakes dataset. Each dot represents a 1/120°x1/120° resolution cell of the regular grid. We will add some clarification in the caption of the figure. We have changed the caption of Figure 5 as in the following:*

***Number of LSWT observations from 1995 to 2022 for Lake Ammalortoq (E) of the ESA CCI LAKES LSWT dataset. Each dot in the plot represents a 1/120°x 1/120° resolution cell of the common regular grid for all the variables of the CCI dataset.***

L119: Please include a reference/DOI to the dataset in the CEDA archive here.

*RESPONSE: OK. Done.*

L125: "detemine" >> "determine"

*RESPONSE: Changed.*

L125: Please define the SST acronym on the previous sentence. Also, you don't need to define the CEDA acronym again here as you have previously defined it on L115

*RESPONSE: We have now defined SST and removed the CEDA acronym.*

L181-184: How has this inference been made? From in situ observations/studies? Please can you provide clarification in the text, perhaps using the papers that you refer to.

*RESPONSE: We have added the following text at the end of the section 2.9.3*

***We have used the temperature of maximum density as a proxy for stratification onset and breakup. However, stratification depends on the lake's depth, size and shape. Some small and shallow lakes wind may not experience thermal stratification since the wind could be strong enough to contrast it and mix the entire lake (Boehrer et al., 2008).  We now further clarify these limitations.***

Figure 2: The text description in L197-199 should be included in the figure caption, along with the exact Landsat 8 and ASTER products shown here.

*RESPONSE: We have moved the text in L197-199 to the Figure 2 caption, and details on the ASTER product used have been added to the Figure 2 caption.*

Figure 3: "…(seeData and Methods)." >> "…(see Data and Methods)."
L219: "Meltwaters" should not be plural. Please change this to "Meltwater" and modify the rest of the sentence to reflect this.

*RESPONSE: Changed*

Figure 4: Specify where the lake area, elevation and minimum observed temperature data is sourced from in the caption.

*RESPONSE: We have added in the caption of Figure 4 the following:*

***Area, albedo and minimum observed brightness temperature have been derived from Landsat 8 observations as described in Sec 2.9.1. The elevation is from the ASTER DEM.***

Figure 4: I think the colour bar for elevation and the y axis for maximum observed temperature should be switched, as intuitively you would associate elevation with the y axis and temperature with the colour bar.

*RESPONSE: The primary purpose of the plot is to look for the influence of albedo on temperature. Albedo is therefore rightly the x-axis and temperature the "dependent"*

*variable plotted in the y-axis. Elevation is provided as a potential confounder of the relationship in colour, and lake size as a potential confounder in dot size. All this seems to us to be the optimum choice.*

L228-230: Please re-word this sentence as I think there are some connecting words missing.

*RESPONSE: This sentence*

**All the six lakes have maximum distance from the closest land (Carrea et al., 2015) less than 2 km which is a useful measure for the size of the lakes in the context of LSWT remote sensing.**

*has now been reworded as in the following:*

**All six lakes have a maximum distance from the closest land (Carrea et al., 2015) of less than 2 km.**

L231: "However, observing these six lakes remain challenging." >> "However, observing these six lakes remains challenging."
*RESPONSE: Changed.*

Table 1: Max. depth and Mean depth rows need a superscript "a" next to them to indicate that they are from the GLOBathy database.
*RESPONSE: Changed.*

L255-256: I think these two sentences between Sections 3.3 and 3.3.1 can be removed as they don't describe any of the results presented in this paper.
*RESPONSE: Done.*

Figure 6. This is a really nice plot effectively showing the seasonal LSWT and LIC results. My only comment is regarding the top panel of the LIC results - The y-axis (labelled % ice) should be a percentage (i.e. 20, 40, 60%) instead of a decimal value (i.e. 0.2, 0.4, 0.6).
*RESPONSE: Done.*

L267: "…even at time of maximum LSWT." >> "…even at the time of maximum LSWT."
*RESPONSE: Done.*

L283: "melt water" >> "meltwater"
*RESPONSE: Done.*

L290: I see you refer to the 2m air temperature as "Tair" since defining this acronym in Table 2. I think you should either define this acronym when the dataset is first introduced (L131) or drop the acronym and refer to it as "2m air temperature" throughout the manuscript.

*RESPONSE: Changed.*

Figure 7: How are the seasonal cycles for each lake location sampled from the ERA5-Land reanalysis dataset? For example, are they the data from the lake centroid position or an average based on a defined lake extent? Can this be added to the figure caption, and perhaps also L175-177.

*RESPONSE: The data are from the centre of the lake (ERA5 Land is at about 9km resolution, regridded at 0.1°). This information has now been added to the Figure 7 caption. Also, in the first sentence of Sec 2.9.2 we have added that the data are from the lakes centre (in bold):*

*The climatology of all variables in this study, except for LIC, has been calculated **at the lake centre** for each day of the year using a 15-day smoothing window (7 days on either side of the central day).*

L321: Can you provide a percentage of ice-free conditions for each monthly lake mean LSWT? And will this include instances where lakes are completely ice-covered? If these are included, surely this will have a marked influence on the monthly lake mean LSWT? I would like to see this better quantified in the text.

*RESPONSE: The number of observations used to compute the monthly lake mean is reported below the LSWT plot for each of the months. In June the number of observations used is less than in the other months shown. This is because sometimes the lake was not ice-free (completely of partially). Please not that the monthly lake mean has been computed on the anomalies and not on the LSWTs directly because anomalies have greater spatial correlation than absolute LSWTs and therefore less data points are needed for a good estimate.*

L342: "…warming of the meltwaters during their passage…" >> "warming of the meltwater during its passage…" ***DONE***
L329-330: Please put brackets around the values and quantified uncertainty. ***DONE***
L369: I am not exactly sure what "meltwater flows" refers to here. Do you mean "meltwater discharge" (i.e. Tair is expected to have a direct influence on the amount of meltwater discharge). ***DONE***
L411: "glacial melting dynamics" >> "glacial melt processes" ***DONE***
L428: "don't" >> "do not" ***DONE***
L435: "Furthermore, lake glacier interaction mechanisms…" >> "Furthermore, lake-glacier interactions and lacustrine processes…" ***DONE***

L439: "behaviors" >> "behaviours" ***DONE***

*RESPONSE: Thank you for spotting these issues. We have made the necessary changes.*

L427-440: This is a powerful statement to make at the end of this manuscript, and absolutely demonstrates the need for in situ observations to ground truth remote

sensing studies in Greenland. What about the wider Arctic? I would like to see Arctic studies brought in here to compare to, particularly those that estimate LSWT from remote sensing (e.g. Dye et al., 2021). I think this study is much more thorough in its analysis than most, and you can effectively demonstrate this with comparison to other studies in the Arctic.

*RESPONSE: Thank you for the suggestion and the reference. We have now added some comments also on ASTER LSWT and reference the paper on the lakes in Sweden (Dye at al 2021).*
*We have added after this sentence:*

*For smaller lakes, higher spatial resolution satellites, such as Landsat 8, can offer observations at a finer spatial resolution (100m) but a much longer revisiting time is often insufficient for a good description of the temporal variations.*

*the following sentence:*

**An example is the use of the high spatial resolution sensor ASTER to characterise the thermal behaviour of lakes in Arctic Sweden where the measurements considered were very sparse in time (Dye et al, 2021).**

*We have also modified the following sentence (modifications in bold):*

*'These are **the** current limitations of the remote sensing data, **which will be partly addressed with improvements in spatial and temporal resolution such as those provided by the satellites for thermal remote sensing such as TRISHNA (Lagouarde et al., 2018) to be launched after 2025**.'*

**References**
Dye, A. et al. (2021) Warm Arctic Proglacial Lakes in the ASTER Surface Temperature Product. Remote Sensing 13(15), 2987. https://doi.org/10.3390/rs13152987

Grinsted, A. et al. (2017) Periodic outburst floods from an ice-dammed lake in East Greenland. Sci Rep 7, 9966 (2017). https://doi.org/10.1038/s41598-017-07960-9

Kjeldsen, K. K. et al. (2017) Ice-dammed lake drainage in west Greenland: Drainage pattern and implications on ice flow and bedrock motion. Geophys. Res. Lett. 44(14), 7320-7327. https://doi.org/10.1002/2017GL074081

Mallalieu, J. et al. (2021) Ice-marginal lakes associated with enhanced recession of the Greenland Ice Sheet. Global and Planetary Change 202, 103503. https://doi.org/10.1016/j.gloplacha.2021.103503

Minowa M., Schaefer M. and Skvarca, P. (2023) Effects of topography on dynamics and mass loss of lake-terminating glaciers in southern Patagonia. Journal of Glaciology. Published online 2023:1-18. doi:10.1017/jog.2023.42

Moon, T. A., M. Fisher, T. Stafford, and A. Thurber (2023). QGreenland (v3.0.0) [dataset], National Snow and Ice Data Center. doi: 10.5281/zenodo.12823307

Sugiyama, S. et al. (2021) Subglacial discharge controls seasonal variations in the thermal structure of a glacial lake in Patagonia. Nat Commun 12, 6301. https://doi.org/10.1038/s41467-021-26578-0

**RESPONSE TO RC2**

The author used remote sensing observational data to analyze the surface water temperature changes of six lakes in West Greenland during the period from 1995 to 2022, aiming to identify the key factors influencing these changes. Overall, the author employed a wealth of remote sensing data, conducted extensive analyses, and drew some interesting conclusions, making this paper promising for publication. I have provided some major comments, and other minor suggestions will be raised during the next round of review to help further improve the paper.

*RESPONSE: Thank you to the reviewer for the constructive comments which will help improving the paper.*

Firstly, the author chose six lakes as the study objects, which represent lakes in the West Greenland region. However, why were only these six lakes selected? How representative are these six lakes? Would it be possible to consider expanding the study to include a larger number of lakes to more comprehensively reflect the trends and characteristics of lake changes in West Greenland? These questions merit further discussion.

*RESPONSE: These lakes were selected since they are large enough to be sensed by the high-quality infrared instruments used for the LSWT and LIC data. These instruments offer high temporal resolution measurements (satellite revisit depends on instrument characteristics and it can be between one and three days) and a spatial resolution of roughly 1km. We did not select more lakes since retrieving LSWT from satellite is challenging given that the size of the lakes in this region is comparable with the satellite resolution. We have selected the six lakes including lakes directly connected to the ice sheet, lakes totally disconnected in land and lakes close to the shore. We have also carried out a spatial characterisation of the lakes in the region using Landsat (a high spatial resolution instrument but with very low revisiting time) to show how representative the six lakes are of lakes in the region.*

We have added some discussion about spatial characterisation of the lakes in the region with high spatial resolution sensor (but low temporal resolution) and put the six lakes in the regional context in section 4.1 where we have specifically added the following:

**The six lakes that have been selected specifically for this study are representative of the lakes in the region exhibiting different albedos, temperatures and elevations. The only limitation to the variety of lakes was the size since meteorological satellites used to create the ESA CCI LAKES dataset have a spatial resolution of about 1km.**

In the introduction section, I suggest adding a detailed review of the existing research progress. This would help readers better understand the current state of research in the field and its gaps, thereby clarifying the potential contribution of the paper. Furthermore, the structure of the introduction could be improved. The current logical arrangement is somewhat unclear, and I recommend reordering it around the core theme of "research significance – research progress – problems to be addressed," making the article more organized and easier to follow.

*RESPONSE: We have conducted a thorough review of the literature, and we have included all the relevant papers on this topic in the introduction. We have added source of insitu data found from literature and the studies that have made use of the data highlighting the limitations of these data.  We have modified the introduction to highlight the need to study in a consistent manner lakes in Greenland using a state-of-the art dataset (the ESA CCI Lakes LSWT and LIC dataset) especially created for climate studies (namely, long-term, consistent and with requirements for quality as defined by GCOS (Global Climate Observing System of the World Meteorological Organisation) to fully characterise thermally lakes (which are representative of the lakes in the region) in Greenland (spatially and temporally) and explore the complex interrelationships between climatic variables and lake dynamics.*

Regarding the Digital Elevation Model (DEM), the author did not use the higher resolution ArcticDEM but instead used other versions of DEM. Could the author consider using the higher resolution ArcticDEM to improve the accuracy of the study results?

*RESPONSE: We used the ASTER DEM for the physical characterisation of the study region which we have conducted using Landsat 8 data. Landsat8 and ASTER data are at the same spatial resolution (30m). We have also found a Greenland specific DEM [Howat, I.M., A. Negrete, B.E. Smith, 2014, The Greenland Ice Mapping Project (GIMP) land classification and surface elevation datasets, The Cryosphere, 8, 1509-1518, doi:10.5194/tc-8-1509-2014] which also has the same resolution and has been derived mainly from the ASTER DEM. Moreover, since Landsat 8 data are at 30m resolution we conclude that higher resolution DEM was not necessary for this study.*

*We have changed the sentence:*

**..., provides a global digital elevation model (DEM) of land surface on Earth that extends from 83°N to 83°S, has at a horizontal spatial resolution of 1 arc second**

*(about 30 metres at the equator). The data are projected onto the 1984 World Geodetic System (WGS84)/1996 Earth Gravitational Model (EGM96) geoid. …*

*Into the following*

*… . It provides a global digital elevation model (DEM) of land surface on Earth that extends from 83°N to 83°S. The data are projected onto the 1984 World Geodetic System (WGS84)/1996 Earth Gravitational Model (EGM96) geoid and the spatial resolution is 1 arc second (about 30 metres at the equator) and they have been resampled at 50 metres resolution to be used with Landsat 8 data for this work. The ASTER GDEM  version 3 dataset …*

Additionally, regarding the lake bathymetry data, the author did not specify its accuracy in the study area. Given the importance of bathymetric data precision for the analysis, could the author further elaborate on the quality of this data and its potential impact on the results?

*RESPONSE: It is very difficult to estimate an error for the bathymetry. In the paper [Khazaei, B., Read, L.K., Casali, M. et al. GLOBathy, the global lakes bathymetry dataset. Sci Data **9**, 36 (2022). https://doi.org/10.1038/s41597-022-01132-9], a validation is carried out for a small portion of the 1.5 million lakes using various metrics and the authors reports a "reasonable accuracy". The lake bathymetry is derived with a model which has been selected among few candidates. The selection is based on the comparison of the predicted maximum depth with the observed value for about 1500 lakes giving NRMSE = 0.17, and ρ = 0.94 for the selected model. Also, a cross validation has been carried out. The actual validation of the bathymetry reported on the paper consists on a visual comparison of the predicted bathymetry with the observed bathymetry for 8 lakes. Regarding the six lakes of this paper, we have found that the maximum depth of one of them is unrealistic and we have pointed it out in the paper. However, currently  this is the best and consistent product for global lake bathymetry.*

*We have changed and added the following text at line 89:*

**GLOBathy provides estimates of bathymetry for lakes worldwide with reasonable accuracy, given the complexity of estimating underwater topography, as reported in the article. The bathymetry is derived with a model which has been selected among few candidates. The selection is based on the comparison of the predicted maximum depth with the observed value for about 1500 lakes giving  the root mean squared error normalized with standard deviation, NRMSE = 0.17, and the Spearman's Rho correlation coefficient , rho=0.94 for the selected model.  Also, a cross validation has been carried out. The actual validation of the bathymetry reported on the paper consists of a visual comparison of the predicted bathymetry with the observed bathymetry for eight lakes. Regarding the six lakes of this paper, we have found that the maximum depth of Lake Eqalussuit Tasiat is unrealistic (see Section 3.2).**

Optical imagery in polar regions is often affected by cloud cover and weather conditions. Has the author taken any effective measures to minimize these influences when selecting the remote sensing data? For example, did the author choose images with less cloud cover or weather disturbance?

*RESPONSE: The LSWT and LIC values have been retrieved only where no clouds where present using algorithms to test for it. For LSWT a quality value reflecting (also) the degree of success of the water-only detection algorithm was used to select the data. The validation of the LSWT through comparison with in situ data distributed globally show a median satellite minus in situ difference of -0.15°C and robust standard deviation of ~0.5°C for the best quality data showing an excellent agreement which we showed that is also stable in time. Furthermore, the retrieval algorithm is physics based and therefore we expect a stable behaviour also for lakes that we cannot directly validate. (see this paper for details: Carrea, L., Crétaux, J., Liu, X., Wu, Y. et al. Satellite-derived multivariate world-wide lake physical variable timeseries for climate studies, Scientific Data, 10, https://doi.org/10.1038/s41597-022-01889-z, 2023.)*

*We have now added the following text at line 108 to summarise the water detection:*

**The lake mask is available at zenodo (Carrea et al, 2022b) and it was derived from the GloboLakes mask (Carrea et al, 2015). For LSWT, the optimal estimation retrieval method of (MacCallum and Merchant, 2012) was applied on image pixels identified as water according to both the lake mask and a reflectance-based water detection scheme (Carrea et al., 2023) which was specifically designed to distinguish water from non-water pixels with clouds, ice or land-contaminated.**

Regarding the reliability of ESA CCI's LSWT and LIC data in Greenland, is there any related validation study or literature supporting their applicability in this region? If no such validation exists, could the author explore the suitability of these data further and address potential sources of error?

*RESPONSE: The ESA CCI LSWT dataset has been created with a physics-based algorithm, and it has been validated using in situ temperature data for lakes distributed globally some of which are at latitude above 55° in environments comparable with those in Greenland. In addition, the LSWT retrieval algorithm has been selected to be based on physics specifically because this gives good reasons to expect stable performance across domains in time and space that cannot be directly easily validated such as Greenland. See details on the ESA CCI LSWT dataset on: Carrea, L., Crétaux, J., Liu, X., Wu, Y. et al. Satellite-derived multivariate world-wide lake physical variable timeseries for climate studies, Scientific Data, 10, https://doi.org/10.1038/s41597-022-01889-z, 2023). The ESA CCI Lakes dataset was specifically created for climate studies according to the requirements of the Global Climate Observing System (GCOS) of the World Meteorological Organisation (WMO).*

*We have added in the paper some information about the dataset and its validation in the Introduction and on the section 2.5 ESA CCI LAKES.*

*We have changed the following sentence starting at line 33*

**Leveraging these capabilities, our study investigates the thermal behaviours of six representative lakes (Figure 1) across West Greenland using satellite-derived time series data of LSWT and Ice Cover (LIC).**

*With*

**Leveraging these capabilities, our study investigates the thermal behaviours of six representative lakes (Figure 1) across West Greenland using satellite-derived time series data of LSWT and Ice Cover (LIC) from the European Space Agency (ESA) Climate Change Initiative (CCI) Lakes dataset specifically created for climate studies according to the requirements of the Global Climate Observing System (GCOS) of the World Meteorological Organisation (WMO) (Buontempo et al., 2022).**

*In addition we have changed from line 116 the following:*

**Because the LSWT retrieval algorithm is based on physics, a stable performance is expected across domains in times and spaces where LSWT cannot be directly easily validated.**

*Into:*

*Because the LSWT retrieval algorithm is based on physics, a stable performance is expected across domains in times and spaces where LSWT cannot be directly easily validated* **through comparison with in situ data, such as for lakes in Greenland. The ESA CCI Lakes dataset has been created as a climate data record following the requirements of GCOS of the WMO (Buontempo et al., 2022).**

The paper utilizes various remote sensing datasets for analysis, but the spatial resolution of these datasets differs significantly. How did the author handle these resolution discrepancies? Were adjustments made to account for differences in the data sources? These aspects should be further elaborated in the paper.

*RESPONSE: This is a very good point. We made adjustments where possible and we considered lake averages rather that considering each of each pixel on the lake in order to draw consistent conclusions. We now include these details in the paper: In section 2.9.2 we have added that we estimate the climatological curves for the lake centre and we have described how we have selected the data from the various datasets; In Section 2.9.4 we have added that we have compared trends and examined correlations on lake means and we described how we extract the data.*

*The following sentence has been added at line 177:*

*Tair and the solar radiation have been extracted at the closest ERA5 resolution cell to the lake centre.*

*We have added the following sentence at line 190:*

*The values of Tair and the solar radiation to be compared with the lake monthly mean have been extracted over a box at the ERA5 resolution cell covering the full lake and have been averaged in space and per month.*

Despite the extensive data analysis conducted, much of the work focuses on data processing and presentation. Could the author add more in-depth discussions to the paper? For example, what are the reasons behind the differences in surface water temperature changes across the lakes? What environmental or climatic factors might be driving these variations? A deeper discussion of these mechanisms would help uncover the underlying causes rather than just presenting the data.

*RESPONSE:*
*We appreciate the suggestion to include more in-depth discussions on the mechanisms driving the observed variations. In response, we have investigated the relationships between lake variables (LSWT and LIC) and atmospheric variables (air temperature and solar radiation). Additionally, for lakes near the coastline, we compared LSWT with the SST of the nearest sea area. Our findings indicate that the largest differences stem from the connection with the ice sheet and the influence of ice melt on the lakes.*

*To address this further, we have conducted a more detailed analysis of these relationships and revised the manuscript to include new subsections in the Discussion section. These subsections focus on:*

1. *Spatial thermal characteristics of lakes in the region with high resolution sensor*
2. *Seasonal trends in LIC and LSWT, providing insights into how these variables evolve throughout the year.*
3. The influence of air temperature and solar radiation on LSWT and stratification, exploring the atmospheric drivers behind these changes.
4. *Temporal variability of LSWT, highlighting patterns and fluctuations over time.*
5. *Implications for LSWT and LIC studies in Greenland and the Arctic, offering broader context and relevance for future research*

*We have also added a conclusion section.*

*These additions aim to uncover the underlying mechanisms driving the observed changes and enhance the interpretative depth of our findings.*

Regarding the structure of the paper, there are some long paragraphs that could make reading difficult for the audience. It is recommended that the author breaks up these sections into shorter paragraphs to improve readability. Furthermore, while the author

presents a large amount of data and discussion, this might dilute the focus of the paper. It is advisable that the author clearly defines the central theme of the paper and structures the analysis around it, which would help readers better understand the paper's key points.

Finally, many of the figures appear too simplistic and at times may confuse the reader. The author should consider improving the design of the figures by adding necessary details and explanations to make them more expressive and easier to interpret. For example, Figure 1 appears to be overly simplified and lacks essential geographical parameters. It would be helpful if the author could add relevant annotations or additional explanations to make the figure more informative and readable.

*RESPONSE:*
*Thank you for your constructive feedback. We appreciate your suggestions and will address each point to improve the clarity and focus of the paper.*

*Paragraph structure: We agree that some paragraphs may be lengthy. We have now broken them into shorter sections to enhance readability and make the text more digestible for the audience.*
*Central theme and focus: We understand the concern regarding the breadth of data and discussion. We have now refined the paper by clearly defining the central theme, ensuring that the analysis is more closely aligned with this theme.*

*Figure design: Thank you for pointing out the simplicity of the figures. We have now revised the figures to include more details and annotations, particularly for Figure 1. We also now add geographical parameters and relevant explanations to ensure the figures are informative and easy to interpret.*

*We appreciate your time and thoughtful suggestions, and we are confident these revisions will improve the overall quality of the paper.*

---

## Author Response (AR2)

**RESPONSES to REVIEWER 1**

Second review for Carrea et al. "Factors influencing lake surface water temperature variability in West Greenland and the role of the ice-sheet"

After partaking in the first round of reviews, the authors of the manuscript ("Factors influencing lake surface water temperature variability in West Greenland and the role of the ice-sheet") have amended and upheld all of the suggested changes. These changes included manuscript re-structuring and introducing Greenlandic lake naming conventions. The manuscript now reads much more clearly, with a distinct structure that guides the reader through the multiple datasets and findings presented.

**Response: the authors thank the reviewer for the very constructive comments which have contributed greatly to improve the paper.**

I only have a three minor comments, but essentially I think this manuscript can be accepted once these changes are implemented.

1. Lengthy sections/paragraphs

I agree with the other first reviewer of this paper that some of the sections/paragraphs are lengthy. This is still the case in this revised version. I don't think this needs much altering other than splitting the following sections into paragraphs:
- 1. Introduction
- 2.1. Study region
- 2.9.1. Characterisation of lakes in the study region
- 3.1. Physical characterisation of the study region (third paragraph)
- 3.2. Characterisation of the six studied lakes (second paragraph)
- 5. Conclusions

To give an example, I think Section 2.1 (Study region) could be split into paragraphs at Lines 82 and 92 to form paragraphs on a) the general region; b) the regional climate; and c) specific lake details.

**Response: We have split the suggested sections in paragraphs.**

2. Revisions to lake names

Thank you for including the Greenlandic lake names in the paper. With regards to how they are referenced throughout the paper; generally, the Greenlandic name references that the placename is associated with a lake (e.g. "Tasia", "Tasersua"), so referring to them as, for example, "Lake Tasersuaq Aallaartagaq (D)" is incorrect. Please amend these like so: "<Greenlandic placename> (Lake <X>)".

For example: "Lake Tasersuaq Aallaartagaq (D)" >> "Tasersuaq Aallaartagaq (Lake D)"

For instances where multiple lakes are referenced, it is fine to write, for example, "The lakes Eqalussuit Tasiat (Lake A), Nassuttuutaata Tasia (B) and Itinnerup Tasersua (C)..." (Line 297-298).

**R: We have followed the structure "<Greenlandic placename> (Lake <X>)", except in list "Eqalussuit Tasiat (Lake A), Nassuttuutaata Tasia (B) and Itinnerup Tasersua (C)..." where we have removed the word 'Lake' in the bracket before the capital letter.**

3. Grammatical errors and typos

There are quite a few grammatical errors and typos remaining, even after a first round of reviews and revisions. I have highlighted those below which I saw, but I also expect all authors to do a final read-through for grammar and spell checking after this round of revisions. For reference, all line suggestions are based on line numbering from the tracked changes version of the revised manuscript.

Line 20: "...such as instance calving..." >> "...such as calving..."
**R: Done**

Line 33-34: "None of the lakes was connected..." >> "None of these lakes are connected..."
**R: Done**

Line 28-40: I would suggest re-structuring this to better convey that actually measurements have been collected as part of three main efforts - 1) the southwest campaign between 2011-2022; 2) the 1998-2000 transect near Kangerlussuaq; and 3) the active, seasonal monitoring of the GEM lakes (I understand that data is only available up until 2019, but they have remained monitoring this lake each summer season and there are plans to publish the 2019-2024 data shortly). As it stands in the current structuring it appears that there are two main campaigns, and the GEM lake monitoring is not a major monitoring effort.
**R: Done. We have created a list of the 3 efforts.**

Line 107: Please add this DOI as a reference, or include the DOI in brackets, rather than as part of the main text.
**R: added the brackets**

Line 118: Again, either add the DOI as a reference, or place it in brackets.
**R: added the brackets**

Line 125-127: I think this sentence needs to be re-ordered, or you can remove "where to discover how to search and download all Landsat products from United States Geological Survey (USGS) data portals)" and move the USGS acronym definition to the next sentence
**R: We have changed the sentence:**

**The Landsat 8 data are available at the Landsat Data Access web page (https://www.usgs.gov/landsat-missions/landsat-data-access) where to discover how to search and download all Landsat products from United States Geological Survey (USGS) data portals. The USGS Landsat no-cost open access data policy remains intact since its inception in 2008.**
into:
**The Landsat 8 data are available at the Landsat Data Access web page (https://www.usgs.gov/landsat-missions/landsat-data-access) within the United States Geological Survey (USGS) data portal. The Landsat no-cost open access data policy of the USGS remains intact since its inception in 2008.**

Line 138: MacCallum and Merchant reference should be inline with the text rather than in brackets, i.e. "(MacCallum and Merchant, 2012)" >> "MacCallum and Merchant (2012)"
**R: Done**

Line 139: "...(Carrea et al., 2023) which was..." >> "...(Carrea et al., 2023), which was..."
**R: Done**

Line 180: DOI as reference or in brackets
**R: added the brackets**

Line 237: "Two domains are addressed: (i) northern domain..." >> "Two domains are addressed: (i) the northern domain..."
**R: Done**

Line 239: "Figure1(b)" >> "Figure 1(b)"
**R: Done**

Figure 2 caption: I think you can remove the "(a), (c)" definitions at the beginning and just have them defined at the end of the sentence. This is the same for the last sentence in the caption defining subsets (b) and (d).
**R: Done, but we have replaced "(a), (c)" with "On the left hand side,....", and "(b), (d)" with "On the right hand side,...."**

Table 1: Please can the lake names caption be amended to acknowledge that lake names are sourced from Oqaasileriffik placename database, which is distributed with QGreenland. I.e. "a. Oqaasilerffik (the Language Secretariat of Greenland) placename dataset, distributed with the QGreenland spatial dataset suite (https://qgreenland.org/) (Moon et al., 2023)
**R: Done but we have removed (the Language Secretariat of Greenland) because of space**

Table 1: "Landsat8" >> "Landsat 8" for captions d and e
**R: Done**

Line 397: Change "lake A" to respective placename

**R: Done**

Line 451-458: Given the manuscript is already lengthy, I don't think this summary passage is needed. I would suggest removing this and going straight into Section 4.1.
**R: Removed the full summary**

Line 550: "...we presented new long-term consistent..." >> "...we presented new, long-term and consistent..."
**R: Done**

Line 589: Initial in reference should be capitalised, i.e. "o." >> "O."
**R: Done**

**RESPONSES to REVIEWER 2**

This is my second review of this paper. The authors have addressed my comments from the previous review, and most issues have been satisfactorily resolved. Even in cases where certain aspects have not been altered, sufficient justifications and corresponding explanations have been provided in the manuscript. Overall, the authors have made significant improvements, and the paper now meets the publication requirements.

Response: **The authors thank the reviewer for the useful suggestions that have definitely improved the paper.**

I still have a few minor suggestions for further enhancement:

1)The authors are commended for citing a large number of relevant references in the introduction. I suggest that these references be appropriately summarized and synthesized, particularly highlighting the research gaps in the related studies to further emphasize the advantages of the current work.

Response: **We have highlighted the current studies that characterise lakes, highlighting their limitations and the need of consistent spatially distributed and long-term measurements to characterise lakes in such a remote area like Greenland. In our opinion, this paper is an important starting point which will allow a better understanding of thermal behaviour of lakes in the region which will allow a better modelling and possibly guide for the selection of crucial area to be monitored.**
**We have also added a paragraph at the end of the Introduction:**
**"In summary, this work aims to provide a substantial initial contribution to understanding lake thermal dynamics in this critical region, with the potential to enhance predictive modelling and support the strategic selection of monitoring locations. Lakes are important sentinels of climate change, as their thermal regimes respond rapidly to atmospheric forcing and reflect broader environmental shifts—particularly in this region. However, consistent in situ observations in this**

area remain limited. Remote sensing offers a powerful tool for capturing spatio-temporal patterns in surface water temperature at otherwise unattainable scales. By addressing this gap and leveraging satellite-based observations, this study also intends to demonstrate the value of remote sensing in supporting climate-related assessments of freshwater systems. Our findings contribute to ongoing efforts to develop robust, observation-driven frameworks for monitoring lake responses to climatic shifts and offer valuable insights for enhancing lake modelling in the region."

2) The paper utilizes a substantial amount of data and describes the data usage process. Would it be possible to include a flowchart to help readers better understand how these data are utilized?

**Response: Given the complexity, rather than a flowchart we have included a table where for each task we have listed the data and task that have been used for. We hope that this table clarifies the data usage.**

3) For the statistical analyses concerning trends and correlations in temperature variations, I recommend including information on confidence levels and related metrics. In particular, how were data gaps in the long-term time series handled?

**Response: Trends are provided with standard uncertainty, and the p-values are used to identify which correlations are significant. We have chosen this approach since, in our opinion, it provides enough information without overcomplicating the matter, given that the paper is already quite dense.**
**The presence of gaps has been handled using anomalies when averaging, which reduces the range of variability of the data. In this way, the averages of data with gaps can be considered representative of the true average when the climatology is added back to the anomalies.**

4) The current method for determining lake stratification employs a fixed temperature threshold. I suggest the authors discuss the rationale behind the choice of this threshold and analyze the sensitivity of the results to its value.

**Response: The temperature of 3.98 °C is the minimum density temperature threshold for freshwater. Deep lakes with temperature seasonally greater than this value are likely stratified for fundamental stability reasons. We have also referenced other studies on lake stratification phenology.**

5) Much of the manuscript is devoted to describing the observed changes, while the underlying mechanisms are less explored. I recommend that the authors, in conjunction with existing literature and theoretical models, delve into the mechanisms driving the changes in lake temperature and other related variables. I know this is not easy, but it's worth trying.

**Response: We thank the author for the comment, and this is something that we are planning to attempt but we also think that this constitutes a separate paper.**

6)I advise standardizing the symbols, units, and legends throughout the paper.

**Response: We have standardized symbols, units and legends.**

7)The language throughout the manuscript should be more concise, avoiding overly long sentences (e.g., Section Introduction). I suggest breaking up some lengthy paragraphs to better highlight the core conclusions and areas for improvement.

**Response: We have followed the suggestion of the reviewers to break up lengthy paragraphs and in particular we have separated into paragraphs the text in the following sections:**
**- 1. Introduction**
**- 2.1. Study region**
**- 2.9.1. Characterisation of lakes in the study region**
**- 3.1. Physical characterisation of the study region**
**- 3.2. Characterisation of the six studied lakes**
**- 5. Conclusions**